# The salivary protein Saglin facilitates efficient midgut colonization of *Anopheles* mosquitoes by malaria parasites

**Dennis Klug** [1]*, **Amandine Gautier**[1], **Eric Calvo**[2], **Eric Marois**[1], **Stéphanie A. Blandin**[1]

**1** Université de Strasbourg, CNRS UPR9022, INSERM U1257, Institut de Biologie Moléculaire et Cellulaire, Strasbourg, France, **2** Laboratory of Malaria and Vector Research, National Institute of Allergy and Infectious Diseases, National Institutes of Health, Rockville, Maryland, United States of America

* dennis.klug@sciencebridge.net

**Data Availability Statement:** All relevant data are included in the manuscript and supporting information files. Transgenic mosquito lines as well as plasmids are available from resource centers (MR4 and Addgene) or upon request from IBiSA

## Abstract

Malaria is caused by the unicellular parasite *Plasmodium* which is transmitted to humans through the bite of infected female *Anopheles* mosquitoes. To initiate sexual reproduction and to infect the midgut of the mosquito, *Plasmodium* gametocytes are able to recognize the intestinal environment after being ingested during blood feeding. A shift in temperature, pH change and the presence of the insect-specific compound xanthurenic acid have been shown to be important stimuli perceived by gametocytes to become activated and proceed to sexual reproduction. Here we report that the salivary protein Saglin, previously proposed to be a receptor for the recognition of salivary glands by sporozoites, facilitates *Plasmodium* colonization of the mosquito midgut, but does not contribute to salivary gland invasion. In mosquito mutants lacking Saglin, *Plasmodium* infection of *Anopheles* females is reduced, resulting in impaired transmission of sporozoites at low infection densities. Interestingly, Saglin can be detected in high amounts in the midgut of mosquitoes after blood ingestion, possibly indicating a previously unknown host-pathogen interaction between Saglin and midgut stages of *Plasmodium*. Furthermore, we were able to show that *saglin* deletion has no fitness cost in laboratory conditions, suggesting this gene would be an interesting target for gene drive approaches.

## Author summary

Female mosquitoes rely on blood feeding to acquire sufficient nutrients for egg development. Because of the importance of this process mosquitoes evolved salivary proteins with a broad range of functions acting as blood thinners, anti-coagulants and immunosuppressants. The effect of these proteins on the blood at the bite site directly influences the size of the blood bolus a female takes up in a given time frame. Both, time of feeding and bolus size, are important parameters for fecundity and survival. Recent studies have shown that a significant proportion of salivated proteins is re-ingested during feeding and becomes part of the blood meal. Here we investigated the salivary protein Saglin which has been previously suggested as putative receptor mediating malaria parasite entry into the salivary

(Infrastructures en Biologie Santé et Agronomie) https://www.ibisa.net/plateformes/insectarium-631.html#mapid, lramolu@unistra.fr.

**Funding:** This work was supported by the Laboratoire d'Excellence (LabEx) ParaFrap (grant LabEx ParaFrap ANR-11-LABX-0024 to SAB), by the Equipement d'Excellence (EquipEx) I2MC (grant ANR-11-EQPX-0022 to EM and SAB), by the ANR grant GDaMo (ANR-19CE35-0007 to EM) and by the ERC Starting Grant Malares (N°260918 to SAB). Additional funding was awarded to DK by the DFG as a postdoctoral fellowship (KL 3251/1-1), and by the Intramural Research Program of NIH/ NIAID to EC (AI001246). The funders had no role in study design, data collection and analysis, decision to publish, or preparation of the manuscript.

**Competing interests:** The authors have declared that no competing interests exist.

gland. By engineering a loss-of-function mutant in *An. coluzzi* we could show that the absence of Saglin impairs the development of parasite stages in the blood meal of the rodent malaria parasite *P. berghei* and the human malaria parasite *P. falciparum* lowering the parasite burden of subsequent stages and preventing efficient transmission at low infection densities. Furthermore, we could show that Saglin is present in the blood meal after feeding possibly indicating a previously overlooked parasite-vector interaction.

## Introduction

Vector borne diseases can be caused by a broad range of different pathogens including viruses (e.g. dengue virus, DENV), bacteria (e.g. *Borrelia*) and unicellular eukaryotes (e.g. *Plasmodium*). Most of these pathogens have in common that successful transmission requires the infection of the salivary glands of a blood feeding vector, often a tick or a mosquito, to be transmitted with the inoculation of saliva during a blood meal. *Plasmodium* parasites are the causative agent of malaria, the most devastating vector borne disease being responsible for more than half a million fatalities each year [1]. *Plasmodium* colonizes the midgut of female *Anopheles* mosquitoes following a blood meal from an infected host. Once ingested, *Plasmodium* gametocytes sense the midgut environment leading to the rupture of the red blood cell membrane upon activation. Subsequently male gametocytes rapidly divide into eight microgametes which engage in active motility to find and fuse with female macrogametes. After gamete fusion a zygote is formed and differentiates within 20–24 h into a motile parasite stage, called ookinete. Ookinetes traverse the midgut epithelial cells to nest between these cells and their basal lamina, facing the hemocoele. In this place ookinetes continue development into oocysts that grow and divide into ~1,500–5,000 crescent shaped sporozoites for 10–12 days [2,3]. This highly motile parasite stage is released from the oocyst into the hemolymph from where the sporozoites invade the salivary glands to be transmitted to a naive host with the next blood meal. Although the precise kinetics of sporozoite invasion into the salivary glands is unknown, it is believed that the time from release into the hemolymph to successful invasion into the salivary gland is only a few minutes [4]. Many sporozoites become stuck in the circulatory system of the mosquito [2], and consequently only 10–20% of midgut sporozoites successfully invade the salivary glands [5,6]. Based on the observation that sporozoites invade exclusively gland tissue, it has been hypothesized that sporozoites possess a specific receptor that recognizes a ligand expressed on the outside of the gland. Indeed, *P. knowlesi* is able to infect and invade the salivary glands of *An. dirus* females while it is unable to invade the salivary glands of *An. freeborni*, although midgut colonization in this species proceeds normally. Strikingly, naive *An. dirus* salivary glands transplanted into *P. knowlesi* infected *An. freeborni* females were successfully invaded whereas conversely, no invasion could be observed into glands of *An. freeborni* transplanted in *An. dirus* [7]. These observations suggest a limited protein interface that is subject to evolutionary selection pressure so that transmission of *Plasmodium* adapts to the mosquito species that is locally suitable as a vector. Still, the knowledge about salivary gland proteins involved in mediating sporozoite recognition and invasion is sparse, and only few proteins have been characterized in any detail. Based on screens probing the interaction of the two abundant sporozoite surface proteins TRAP and CSP with proteins resident on the salivary glands, Saglin, SG1-like protein and the CSP binding protein (CSP-BP) have been identified as putative ligands for sporozoite entry into salivary glands [8, 9]. The SG1-like protein, considered similar to SG1 proteins despite a major difference in size, belongs to a family of proteins with high molecular weight (SGS family) and is conserved across *Anopheles*, *Culex*

and *Aedes*. In accordance with these observations a member of the SGS family in *Aedes aegypti*, aaSGS1, was shown to play a role in the salivary gland invasion process of *P. gallinaceum* sporozoites [10]. However, a recent study on aaSGS1 using knockout mosquitoes has shown that this protein mainly affects the colonization of the midgut by *P. gallinaceum* rather than the invasion of sporozoites into the salivary glands [11]. Another interesting protein potentially involved in sporozoite invasion into the salivary glands is CSP-BP, which is highly conserved across insects and corresponds to UPF3 in *Drosophila*. UPF3 belongs to a family of RNA binding proteins which are implicated in mediating quality control of mRNAs [12, 13]. How CSP-BP, which is believed to function intracellularly, mediates sporozoite invasion still needs to be investigated. By far the best characterized salivary gland protein thought to mediate *Plasmodium* sporozoite recognition and invasion is Saglin. Saglin was identified by mass spectrometry as a component of *Anopheles* saliva and has been biochemically characterized as a 50 kDa protein forming a disulphide bonded homodimer [14]. Interestingly, ingestion of a monoclonal antibody targeting Saglin inhibited the salivary gland invasion rate of *P. yoelii* sporozoites by 73% [15] while intrathoracic injection of the same antibody blocked invasion of *P. falciparum* sporozoites by 95% [9]. The circular peptide SM1 was identified by multiple rounds of injections and enrichment of a phage display peptide library in mosquitoes to discover salivary gland and midgut ligands [16]. Like the sporozoite protein TRAP, SM1 interacts with mosquito Saglin, and its presence blocks sporozoite colonization of the gland, leading to the model that recognition of the salivary gland by TRAP is a prerequisite for sporozoite invasion [9,16]. Accordingly, the RNAi mediated knockdown of Saglin has also been shown to reduce the efficiency of sporozoite penetration into the salivary glands [9]. However, artificial overexpression of Saglin in the distal-lateral lobes of the salivary gland failed to boost sporozoite loads, raising doubts about the function of Saglin in sporozoite invasion [17]. In *An. gambiae*, the SG1 family, of which Saglin is a member, consists of seven small salivary gland proteins with unknown functions, all of which are specifically expressed in females [18]. In line with this observation, most SG1 genes are encoded on the X-chromosome, while only one member, *AgTRIO*, is found on an autosome (2R) (**Fig 1A and 1B**). Although all proteins in the family are similar in length, the degree of overall sequence conservation is low (**S1 Fig**). Recently the expression pattern of *trio* and *saglin* promoters has been investigated by creating reporter lines expressing fluorescent proteins [19]. Both *trio* and *saglin* promoters drive expression exclusively in the median lobe of the female salivary gland, but their fine expression patterns are different (**Fig 1A**). In addition, the intergenic 5' sequence upstream of the *saglin* gene appears to lack intrinsic transcriptional activity, suggesting that the closely clustered X-linked SG1 genes likely share common transcriptional elements and a similar expression pattern [19]. Beside Saglin, the SG1 protein AgTRIO has also been shown to affect sporozoite behaviour. Indeed, knock-down of *AgTRIO* in *P. berghei* infected mosquitoes negatively affects sporozoite transmission [20], and administration of α-AgTRIO antibodies to mice significantly reduced infectivity of sporozoites in these mice [21]. Still, the molecular mechanisms underlying SG1 protein effect on sporozoite biology remain largely unknown. This study aims at better understanding how Saglin affects *Plasmodium* development in the mosquito.

## Material & methods

### Ethics statement

Experiments were carried out in conformity with the 2010/63/EU directive of the European Parliament on the protection of animals used for scientific purposes. Our animal care facility received agreement #I-67-482-2 from the veterinary services of the département du Bas Rhin (Direction Départementale de la Protection des Populations). The use of animals for this

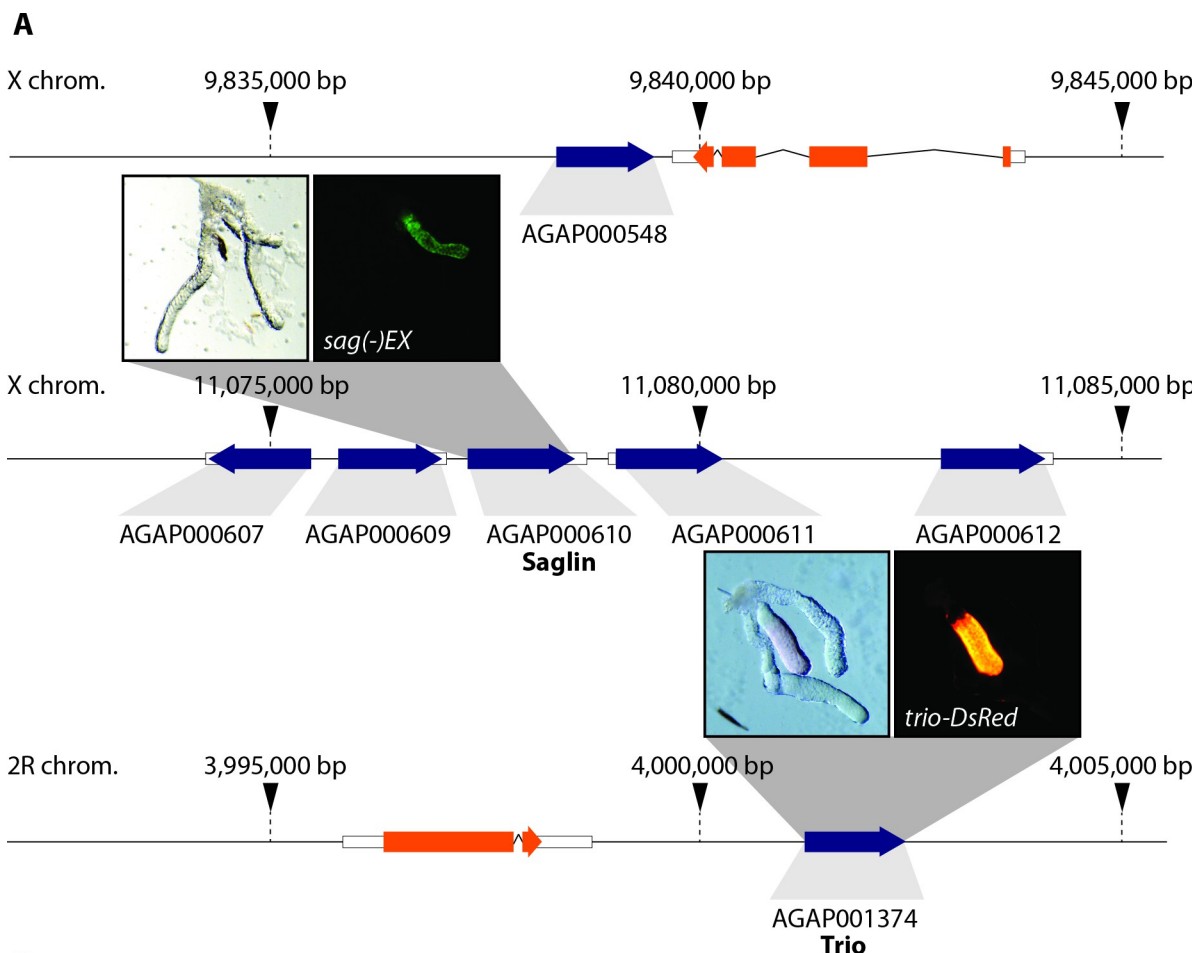

**Fig 1. SG1 family proteins in the *An. gambiae* complex. A)** Genomic organization of the SG1 family according to VectorBase (Release 54). Genes encoding SG1 proteins are shown in blue and unrelated genes in orange. Note the high proportion of X-linked genes and the absence of introns in the entire protein family. Representative images of salivary glands dissected from a *sag(-)EX* and a *trio-DsRed* female where EGFP and DsRed are expressed under the control of the *saglin* and *trio* promoters. Both promoters are active in the median lobe. B) Table summarizing different parameters and the presence of specific domains for all members of the SG1 family. Sequence information was retrieved from VectorBase (version 54) [30]. Signal peptide (SP) predictions were performed using SignalP 5.0 [31], transmembrane domains (TMD) were predicted with TMHMM 2.0 [53], GPI-anchors (GPI) with GPIpred [54] and other known domains with SMART [32].

**B**

| Gene ID | Length (bp) | Length (aa) | TMD | SP | GPI | Known domains | Chrom. |
|---|---|---|---|---|---|---|---|
| AGAP000548 | 1.158 | 385 | No | Yes | No | None | X |
| AGAP000607 | 1.179 | 392 | No | Yes | No | None | X |
| AGAP000609 | 1.206 | 401 | No | Yes | No | None | X |
| AGAP000610 | 1.296 | 431 | No | Yes | No | None | X |
| AGAP000611 | 1.248 | 415 | No | No | No | None | X |
| AGAP000612 | 1.206 | 401 | No | Yes | No | None | X |
| AGAP001374 | 1.176 | 391 | No | Yes | No | None | 2R |

project, and the generation and use of transgenic lines (bacteria, mosquitoes, parasite) were authorized by the French ministry of higher education, research and innovation under the agreements number APAFIS#20562–2019050313288887 v3, and number 3243, respectively.

## Animals

Knock-in transgenesis was performed in a *vasa-Cas9* expressing *An. coluzzii* strain which had been introgressed into the *Ngousso* genetic background [22]. As a positive control of *GFP* expression in the midgut we used the *An. stephensi* line G12-GFP [23]. Male CD-1 mice (purchased from Janvier Labs and further self-bred at the in-house facility) were used to perform *P. berghei* infections and female CD-1 mice were used for routine blood feedings to propagate mosquito colonies.

## Mosquito breeding

*An. coluzzii* and *An. stephensi* mosquitoes were reared in standard conditions (27°C, 75–80% humidity, 12-hr/12-hr light/dark cycle) as described previously [33]. In brief, larvae were reared in plastic pans filled with water purified by osmosis and fed daily with pulverized fish food (TetraMin). Pupae were collected in small glass dishes and transferred into netted cages. Hatched mosquitoes were fed on 10% sugar solution *ad libitum*. To propagate colonies, ≥4 day old mosquitoes were blood fed for 10–15 minutes on anesthetized mice. Two-three days later a glass dish with wet filter paper was provided to allow egg laying. Larvae hatched approximately 48 h after the deposition of eggs.

## Estimating fitness of the *sag(-)KI* transgene by flow cytometry

A Complex Object Parametric Analyzer and Sorter (COPAS, Union Biometrica) was used as described [34,35] to quantify the frequency of the *sag(-)KI* transgene in mixed populations of neonate mosquito larvae and to count and mix defined numbers of homozygous *sag(-)KI* and wild-type larvae for experiments. Flow cytometry was always performed on unfed first instar larvae in water free of any debris to minimise background fluorescence and ensure accurate sorting. To assess the fitness of the *sag(-)KI* transgene, a mosquito colony was started by crossing 24 homozygous *sag(-)KI* females with 24 wild-type males. Subsequently the frequency of *sag(-)KI* carriers and non-carriers was measured by COPAS sorting of neonate larvae for 10 generations. Flow cytometric data were analyzed using FlowJo v10.8.1. and ratios of transgene carriers and non-carriers were plotted using GraphPad Prism 5.0.

## Infections with *Plasmodium berghei*

To infect mosquitoes with the rodent malaria parasite *P. berghei*, male CD1 mice were injected intraperitoneally with 200 µl of thawed mouse blood containing *P. berghei* blood stages. Infections were performed with the *P. berghei* line Δp230p-GFP [24] genetically engineered to constitutively express *EGFP* under the control of the promoter of heat shock protein 70 (*hsp70*). Upon infection parasitemia was monitored by flow cytometry (AccuriC6 SORP, Becton Dickinson). Once parasitemia reached ≥1%, mice were bled by cardiac puncture and blood stage parasites were transferred into naïve mice by intravenous injection into the tail vein. Typically, $2 \times 10^7$ parasites were transferred per mouse, leading to a parasitemia of 1.5–3.5% three days post injection when mosquitoes were allowed to feed on the anaesthetized mice (same conditions as for regular blood feeding). For standard/high infections, mice were used three days after passage. For lower infections, mice were used two days after passage. Parasites were cycled regularly through mosquitoes to maintain fitness. Parasite stocks were stored at -80°C by mixing equal volumes of infected blood with a parasitemia of ≥1% and phosphate buffered saline (PBS) supplemented with 30% glycerol.

## Culture of *P. falciparum* and mosquito infection

Gametocytes were produced *in vitro* using the *Plasmodium falciparum* NF54 clone A11 [25,26] and similar procedures as in [36]. All cultures were maintained in A+ human red blood cells (EFS, Strasbourg) at 3.8% haematocrit and complete medium (RPMI 1640 with L-glutamine and 25 mM HEPES supplemented with 10% human A+ serum (EFS, Strasbourg) and 10 mM hypoxanthine (c-c-Pro, Oberdorla), at 37°C, under a 5% $O_2$, 5% $CO_2$ and 90% $N_2$ atmosphere. Briefly, asexual parasite cultures were kept at 3% parasitemia maximum. Gametocyte cultures were seeded in a 6-well plate at 0.5% parasitemia and maintained in culture with daily medium changes for 16 days before infection. The quality and density of gametocytes was regularly evaluated on blood smears. Gametocyte infected RBCs were mixed with non-infected RBCs and human serum (50% hematocrit, 0.6–1.5 million gametocytes stage V per mL of reconstituted blood) and offered to starved female mosquitoes using a Hemotek system mounted with Parafilm to maintain the blood meal at 37°C. Unfed females were discarded 1-3h post feeding, and the remaining maintained at 27°C, 70% humidity for 7–8 days before dissection.

## Parasite counting in mosquito midguts and salivary glands

To assess parasite burdens, infected mosquitoes were anesthetized on ice and transferred into a Petri dish containing 70% ethanol. After 1–2 minutes mosquitoes were transferred into a second Petri dish containing phosphate buffered saline (PBS). For oocyst counting, mosquitoes were dissected 7–8 days post infection (dpi) on a microscope slide in a drop of PBS using a SZM-2 Zoom Trinocular stereomicroscope (Optika). Dissected midguts infected by GFP-expressing *P. berghei* were transferred on a second microscope slide in a drop of fresh PBS, covered with a cover slip and imaged using a Nikon SMZ18 fluorescence stereomicroscope. Images were then processed using the watershed segmentation plugin [37] and oocysts were subsequently counted using the „analyze particles"function implemented in Fiji [29]. Midguts infected by *P. falciparum* were dissected in PBS and stained with 0.2% mercurochrome in water. Samples were washed three times in PBS for 10 min each before imaging using a Zeiss Axio Zoom.v16 microscope. *P. falciparum* oocysts were counted manually using the "multi-point" tool of Fiji [29].

For sporozoite counting, pools of salivary glands from ≥10 mosquitoes (17–18 dpi) were dissected and manually ground in 50 μl PBS with a plastic pestle for one minute in a 1.5 ml plastic reaction tube. Samples were diluted by adding 50 μl of PBS used to carefully rinse the pestle. Subsequently 7 μl of sample solution were loaded into a Neubauer hemocytometer. Sporozoites were allowed to settle for 5 minutes and counted using a light microscope (Leica DFC3000 G) with 20-fold magnification.

## Analysis of transmission capability *in vivo*

Freshly hatched *sag(-)KI* larvae obtained from a homozygous colony were mixed in equal amounts with wild-type larvae (*Ngousso*), raised together and infected with the GFP-expressing *P. berghei* as described above. Blood fed females were collected and kept at 21°C and 60–70% humidity. Seven days after blood feeding, midguts from a random sample of mosquitoes were dissected to monitor parasite load and prevalence. On the 15th day after infection mosquitoes were briefly anaesthetized on ice and sorted as wild-type or *sag(-)KI* females according to their EGFP expression in the eye (typical expression pattern of the 3xP3 promoter). Subsequently groups of 10 mosquitoes of identical genotype were distributed in paper cups. On the evening of the 17th day after infection sugar pads on cups were removed to starve mosquitoes overnight. The following day one anaesthetized mouse was placed on each cup and mosquitoes were allowed to blood feed for 15 minutes. A minimum of three and a maximum of eight

mosquitoes per cup were observed to have successfully taken blood. Parasitemia in mice was monitored by flow cytometry (AccuriC6 SORP, Becton Dickinson) from day three to day seven after biting. On day seven, all infected animals were sacrificed to avoid suffering and non-infected mice followed for an additional week. All transmission experiments were performed with male CD1 mice with an approximate age of six months.

## Analysis of blood feeding behaviour

Neonate homozygous *sag(-)KI* and wild-type larvae of the same colony were either COPAS sorted or obtained from homozygous *sag(-)KI* and control *Ngousso* colonies, and raised as a mix. Adult females aged ≥4 days were allowed to blood feed for 10 minutes. After blood feeding females were anaesthetized on ice and sorted to discard unfed females and, if necessary, to count GFP-positive and GFP-negative individuals to calculate the feeding efficiency for both genotypes. Blood fed females were shock frozen for 10 minutes at -20˚C and subsequently weighed individually using a fine scale (Ohaus, Explorer Pro). As a control, females that were kept under similar conditions but had not yet received a blood meal, were weighed. Pools of 25–50 females were transferred to 1.5 mL tubes and weighed as previously described. The average individual weight was calculated by dividing the total weight by the number of mosquitoes. To calculate the number of ingested cells, blood fed females were kept unfrozen. Their abdomens were dissected, transferred into a tube containing either 1,500 μl or 2,000 μl PBS and crushed with a pestle to release the ingested blood. Seven μl of the solution were loaded in a Neubauer counting chamber. Red blood cells were allowed to settle for 5 min before counting.

## Rescue experiments

Mice were infected with *P. berghei* and anaesthetized as described above. Subsequently mice were injected intravenously with (1) 100 μl PBS, (2) a mixture of 50 μl recombinant Saglin (rSag, 1 mg/ml in 25 mM phosphate buffer, 500 mM NaCl, pH 7.4) + 50 μl PBS, or (3) 50 μl of PBS + 50 μl foetal bovine serum containing 50 dissected salivary glands (equivalent of 25 mosquitoes) of wild-type or *sag(-)KI* females. After approximately 10 minutes mice were exposed to *sag(-)KI* and wild-type mosquitoes, which were evaluated for parasite burden in the midgut 7–8 days later. Salivary gland extracts were prepared in advance by dissecting 50 salivary glands from mosquitoes in 50 μl fetal bovine serum. Samples were subsequently shock-frozen on dry ice and stored until use at -80˚C. Depending on the condition, samples were completed to 100 μl by adding PBS immediately before injection.

## Exflagellation assay

Four to ten-day old *sag(-)KI* and wild-type mosquitoes were allowed to take a blood meal on an infected mouse for five minutes. Mosquitoes were then kept at 21˚C and 60–70% humidity for 12 minutes to activate gametocytes and induce exflagellation of male gametes, briefly anaesthetized on ice and their midguts were opened to release and smear the complete blood meal on a microcopy slide using two syringes. To compensate for possible differences in exflagellation due to time differences resulting from the generation of blood smears, five blood smears of wild type and *sag(-)KI* were always alternated. Blood smears were air dried, fixed for one minute in methanol and stained using Diff-Quik (RAL Diagnostics). Male gametes were counted using a light microscope (Leica DFC3000 G) with 100-fold magnification and a counting grid. For each sample, microgametes were counted in 30 fields, as well as the number of red blood cells in one representative field. Microgametocytemia was calculated as follows: (total number of exflagellated microgametes in 30 fields) / ((number of RBCs in one representative field) x 30) x 100.

### Fluorescence imaging

Imaging of DsRed or EGFP in mosquitoes, midguts and salivary glands was performed using a Nikon SMZ18 stereomicroscope with a Lumencor Sola Light engine, in brightfield and with appropriate fluorescence filters. Scale bars were implemented in reference to an objective micrometer (Edmund optics) that was imaged with the same magnification. Images were edited using Fiji [29].

### Expression of recombinant Saglin

The *saglin* gene (AGAP000610, coding for mature peptide only) was codon-optimized for *E. coli* expression and subcloned into pET19b by Biobasic Inc., Markham, Canada. The synthetic gene was designed to contain *NdeI* (5'-end) and *XhoI* (3'end) restriction sites and a 6xHis-tag before the stop codon. Expression of rSag in *E. coli* (BL21pLYS) was performed under standard conditions as described previously [38]. Purification of rSag was carried out by Immobilized metal affinity chromatography (IMAC) followed by Size-exclusion chromatography (SEC) using the Akta Purifier system (GE Healthcare, Piscataway, NJ). Briefly, cell lysates containing the recombinant protein were supplemented with 500 mM NaCl and 5 mM imidazole and loaded onto a HiTrap Chelating HP column charged with NiCl2 (5 mL bed volume, GE Healthcare, Piscataway, NJ, USA). Fractions were eluted using a linear gradient 0–1 M Imidazole in 25 mM phosphate buffer, 500 mM NaCl, pH 7.4 over 60 min at 2 ml/min. Fractions containing rSag were pooled and concentrated using Amicon Ultra-15 centrifugal filter units (Millipore Sigma) and fractionated on a Superdex 200 10/300 GL column (GE Healthcare, Piscataway, NJ). Purity was verified by NuPAGE 4–12%, Bis-tris (ThermoFisher Scientific), visualized by Coomassie staining. Protein identity was confirmed by N-terminal sequencing using automated Edman degradation (Research Technology Branch, NIAID).

### Generation of α-Saglin antibodies

Polyclonal antibodies against rSag were raised in rabbits as described previously [39]. Briefly, immunization of rabbits (New Zealand White) was carried out in Noble Life Science facility (Woodbine, MD) according to their standard protocol. Rabbits received a total of 3 immunizations with 1 µg of recombinant protein each at days 0, 21, 42. Freund's Complete Adjuvant was used for the initial immunization and Freund's Incomplete Adjuvant was utilized for the subsequent boosters. Rabbit sera were collected by exsanguination 2 weeks after the last boost.

### Western blotting of different mosquito tissues

Ten Salivary glands (equivalent of five mosquitoes) or five midguts were dissected in 20 µl Laemmli buffer. Hemolymph collection from 20 female mosquitoes was performed by proboscis clipping directly into 10 µl Laemmli buffer. To collect proboscises, mosquitoes were anaesthetized on ice, 20 proboscises (equivalent of 20 mosquitoes) were cut using scissors and transferred into 20 µl Laemmli buffer using forceps. To assess distribution of Saglin in the mosquito body, four female mosquitoes with salivary glands removed were collected in 100 µl Laemmli buffer. Salivary glands, midguts, proboscises and carcasses were crushed manually using pestles for 30–60 sec before samples were denatured at 65˚C for 5 minutes in a heat block. A stock of rSag (1mg / ml) was diluted 1:20 in Laemmli buffer and denatured together with other samples as positive control. Subsequently samples were centrifuged for 3 min at 13,000 rpm and a maximum of 20 µl per sample were loaded on a Mini-PROTEAN TGX Stain-Free Precast Gel (BioRad) using 6 µL of Thermo Scientific PageRuler Plus Prestained Protein Ladder as standard. Separation was performed at 170 V using the Mini Trans-Blot cell

system (BioRad). Once separation was complete, gels were blotted on a PVDF membrane (Trans-Blot Turbo Mini 0.2 μm PVDF Transfer Pack; BioRad) using the mid-range program of a Pierce Fast blotter (Thermo Fisher Scientific). The membrane was blocked for one hour in PBST (PBS containing 0.1% Tween) supplemented with 5% fat-free milk powder. Membranes were incubated overnight at 4˚C with primary antibody diluted in PBST supplemented with 3% fat-free milk powder (PBST3M). Blots were washed three times for 10 minutes in PBST and incubated for one hour at room temperature in secondary antibody conjugated to horse-radish peroxidase (HRP) diluted in PBST3M. After incubation with secondary antibodies, membranes were washed three times for 10 minutes with PBS. Antibody binding was revealed using the Super signal WestPico Plus kit (Thermo Fisher Scientific). After 1–2 minutes of incubation, images were acquired using the Chemidoc software (Biorad). Before incubation with further primary antibodies to visualize additional proteins, membranes were stripped for 20 to 30 minutes in Restore PLUS Western Blot Stripping Buffer (Thermo Fisher Scientific) and washed three times for 10 minutes in PBST followed by a new incubation in blocking solution.

## Western blotting and Coomassie staining of mosquito midgut contents

Mosquitoes were allowed to feed blood on an anaesthetized BALB/c mouse for a maximum of 5 minutes. Subsequently mosquitoes were anaesthetized on ice and midguts of five blood or sugar-fed siblings were dissected on a microscopy slide. In preparation of the dissection a 20 μl drop of PBS premixed with heparin was placed next to each mosquito destined for dissection. Removed midguts were placed within the drop before opening. Subsequently midgut contents of five guts were collected with a pipette and transferred to a fresh plastic reaction tube yielding an approximate volume of 100 μl. To avoid degradation of proteins within the blood meal, dissections were performed within 10 minutes after blood feeding. Samples were homogenized by passing through an insulin syringe for 10 times and clarified by centrifugation for 1–2 min at 14,000 g and 4˚C. Subsequently protein concentration was measured (Denovix) and samples were stored at -80˚C before use. For electrophoresis 50–75μg of protein per sample were loaded on a Nu-PAGE and separated using MES buffer. Samples were run in duplicate to allow for Coomassie staining and immunoblotting. Membrane transfer was performed using iBLOT 2 (Invitrogen) overnight at 4˚C in TBS-T buffer supplemented with 5% milk powder. Immunoblots were incubated with α-Saglin antibodies (1:800) in TBS-T supplemented with 0.05% milk powder, washed four times in TBS-T and incubated with secondary antibody (1:10,000). Immunoblots were developed with the Super signal WestPico Plus kit (Thermo Fisher Scientific).

## Indirect immunofluorescence staining of salivary glands

Immunofluorescence staining of Saglin expression in salivary glands was performed according to [17]. The protocol was modified to treat the samples with PBSBT (PBS supplemented with 1% BSA and 0.1% Triton X-100) only once for 5 minutes and then three times for 10 minutes after treatment with the secondary antibody (Table 1). The primary α-Saglin antibody and the secondary Alexa Fluor 546 coupled anti-rabbit antibody were both diluted 1:200. Imaging was performed using a Zeiss LSM 780 microscope equipped with a Hamamatsu Orca Flash 4.0 V1 camera using a 63x (NA 1.4) objective.

## Injection of fluorescently labeled Wheat Germ Agglutinin (WGA) in mosquitoes

Mosquitoes 4–10 days post emergence were injected twice using the Nanoject III Injector (Drummond) to reach a total of 100 nl of XFD594 (structural analogue of Alexa Fluor 594)

**Table 1. Key Resources.**

| Reagent type(species) or resource | Designation | Source or reference | Identifiers | Additional information |
|---|---|---|---|---|
| Strain, (*An. coluzzii*) | *Ngousso* | Morlais Isabelle, IRD/OCEAC, Yaounde | MRA-1279 | Wild-type strain |
| Strain, (*An. coluzzii*) | *aapp-DsRed* | [19] | / | DsRed expression in distal-lateral lobes of the salivary gland |
| Strain, (*An. coluzzii*) | *trio-DsRed* | [19] | / | DsRed expression in median lobe of the salivary gland |
| Strain, (*An. coluzzii*) | *sag(-)KI* | [19] | / | Knockin into *saglin* gene expressing EGFP from 3xP3 and endogenous *Saglin* promoters |
| Strain, (*An. coluzzii*) | *sag(-)EX* | [19] | / | Knockin into *saglin* with excised 3xP3 promoter expressing EGFP from endogenous *saglin* promoter |
| Strain, (*An. coluzzi*) | *vasa-Cas9* | [22] | / | *Cas9* expression in the germline by the *vasa* promoter |
| Strain, (*An. stephensi*) | *G12-GFP* | [23] | / | *GFP* expression in the midgut by the *G12* promoter |
| Strain, (*Mus musculus*) | *CD-1* | Janvier Labs, Le Genest-Saint-Isle, France | / | self-bred in in-house facility |
| Strain, (*Plasmodium berghei*) | *Δp230p-GFP* | [24] | / | Constitutive expression of GFP along the life cycle |
| Strain, (*Plasmodium falciparum*) | NF54 | [25, 26] | / | / |
| Protein | recombinant Saglin | this study | / | / |
| Antibody | α-PPO2 | [27] | / | Rabbit polyclonal |
| Antibody | α-Saglin | this study | / | Rabbit polyclonal |
| Antibody | α-GFP | ThermoFisher Scientific, Waltham, US | Cat#PA1-980A | Rabbit polyclonal |
| Antibody | α-mCherry | abcam, Cambridge, UK | Cat#167453 | Rabbit polyclonal |
| Antibody | α-TEP1 | [28] | / | Rabbit polyclonal |
| Antibody | Antibody coupled to HRP | Promega, Madison, US | Cat#W4011 | Goat anti rabbit secondary antibody |
| Antibody | Antibody coupled to Alexa Fluor 546 | ThermoFisher Scientific, Waltham, US | Cat#A-11035 | Goat anti rabbit secondary antibody |
| Software, algorithm | Prism 5.0 | GraphPad, San Diego, US | / | https://www.graphpad.com/scientific-software/prism/ |
| Software, algorithm | FlowJo v10.8.1. | BD Biosciences, Ashland, US | / | https://www.flowjo.com/ |
| Software, algorithm | FIJI | [29] | / | https://imagej.net/software/fiji/ |
| Software, algorithm | MetaMorph | Molecular Devices, San Jose, US | / | https://www.moleculardevices.com/contact |
| Software, algorithm | VectorBase (version 36–55) | [30] | / | https://vectorbase.org/vectorbase/app |
| Software, algorithm | SignalP 5.0 | [31] | / | https://services.healthtech.dtu.dk/service.php?SignalP-5.0 |
| Software, algorithm | SMART | [32] | / | http://smart.embl-heidelberg.de/ |
| Software, algorithm | NIS-Elements, F 4.20.01, 64 Bit | Nikon Instruments Inc., Melville, US | / | https://www.microscope.healthcare.nikon.com/products/software/nis-elements |
| Other | WGA-XFD594 | AAT Bioquest, Sunnyvale, US | Cat#25509 | / |
| Other | Hoechst 33342 | ThermoFisher Scientific, Waltham, US | Cat#H3570 | / |
| Other | Ponceau S solution | Sigma-Aldrich, St Louis, US | Cat#P7170-1L | / |
| Other | Diff-Quik Solution I and II | RAL Diagnostics, Martillac, France | Cat#10736133 Cat#10736134 | / |

labeled WGA at a concentration of 40 μg/ml. Salivary glands were dissected one hour after injection and imaged with a Zeiss LSM 780 confocal microscope equipped with a Hamamatsu Orca Flash 4.0 V1 camera using a 40x (NA 1.4) objective. Fluorescence patterns were quantified using Fiji [29].

### Statistical analysis

Statistical analysis was performed using GraphPad Prism 5.0 (GraphPad, San Diego, CA, USA). Depending on data sets statistics were performed with Mann Whitney test, one-way ANOVA or Fisher's exact test. A value of $p < 0.05$ was considered significant.

## Results

### Absence of Saglin impairs midgut colonization by *Plasmodium berghei* and *Plasmodium falciparum*

To investigate the impact of the loss of Saglin expression on the susceptibility of mosquitoes to *Plasmodium* infection, we made use of the mosquito line *sag(-)KI* in which the coding sequence of Saglin (AGAP000610) is replaced with EGFP [19] (**Fig 2A**). In a first experimental setup, homozygous *sag(-)KI* mosquitoes and wild-type siblings obtained from the same colony were co-cultured and infected together with the rodent malaria parasite *P. berghei*. Determination of parasite loads in midguts dissected on days 7 and 8 post infection revealed a median of 16 oocysts (mean: 31 oocysts) in *sag(-)KI* females, whereas wild-type females were infested with a median of 88 oocysts (mean: 122 oocysts) indicating a significant difference in the susceptibility to *Plasmodium* infection in *sag(-)KI* females (**Fig 2B**). To simplify mosquito breeding for subsequent infection experiments, we then bred wild-type and homozygous *sag(-)KI* mosquitoes as separate colonies, and neonate larvae of both colonies were either mixed or reared separately, but always synchronously to achieve equal proportions of males and females of both genotypes. To test for possible biases induced by the new breeding scheme, infection experiments were repeated. Infected *sag(-)KI* and wild-type mosquitoes showed a median of 11 oocysts (mean: 29 oocysts) and of 32 oocysts (mean: 56 oocysts) respectively, confirming the phenotype (**S2 Fig**). Despite the drop in oocyst numbers observed in *sag(-)KI* females, prevalence for infection was not significantly different from wild-type, even if prevalence in the *sag(-)KI* population was reproducibly slightly lower compared to the wild-type (**Figs 2B and S2**). To further test if the observed changes in the midgut parasite burden result in lower sporozoite concentrations in the salivary glands, we counted salivary gland sporozoites (SGS) at days 17–18 post infection. SGS numbers in *sag(-)KI* females were indeed significantly reduced (**Fig 2C**). While wild-type females displayed a median of 30,000 SGS (mean: 30,400 SGS) per mosquito, a median of 6,700 SGS (mean: 7,100 SGS) per mosquito was observed in *sag(-)KI* females (>75% reduction in sporozoite load) (**Fig 2C**). Based on this observation, we investigated whether the reduction in SGS was solely due to reduced oocyst numbers, or whether sporozoites in *sag(-)KI* mosquitoes also exhibit impaired salivary gland recognition, as Saglin had been previously described as a determinant of salivary gland entry [9]. We quantified the sporozoites invasion rates in the salivary glands of *sag(-)KI* and wild-type mosquitoes by calculating the ratio of SGS per oocyst based on results obtained from the same batches of infected mosquitoes. Means of 347 and 162 SGS per oocyst (median: 175 and 170 SGS per oocyst) were obtained for *sag(-)KI* and wild-type females, respectively, indicating that recognition of the salivary glands by sporozoites in the absence of Saglin is not impaired (**Fig 2D**). We further tested whether *sag(-)KI* would also affect mosquito susceptibility to the human malaria parasite *P. falciparum*. While 65% of wild-type mosquitoes carried parasites, only 13% of *Sag(-)KI* mutants were infected, and those infected carried fewer parasites than wild-type (median of 2 vs. 7, respectively) (**Fig 2E**). The reduced susceptibility of *Sag(-)KI* mosquitoes to *P. falciparum* was further confirmed in a second experiment with higher infection levels: prevalence of 76% vs. 92%, and median load of 11 vs. 35 oocysts per midgut in *Sag(-)KI* vs wild-type, respectively (**Fig 2E**).

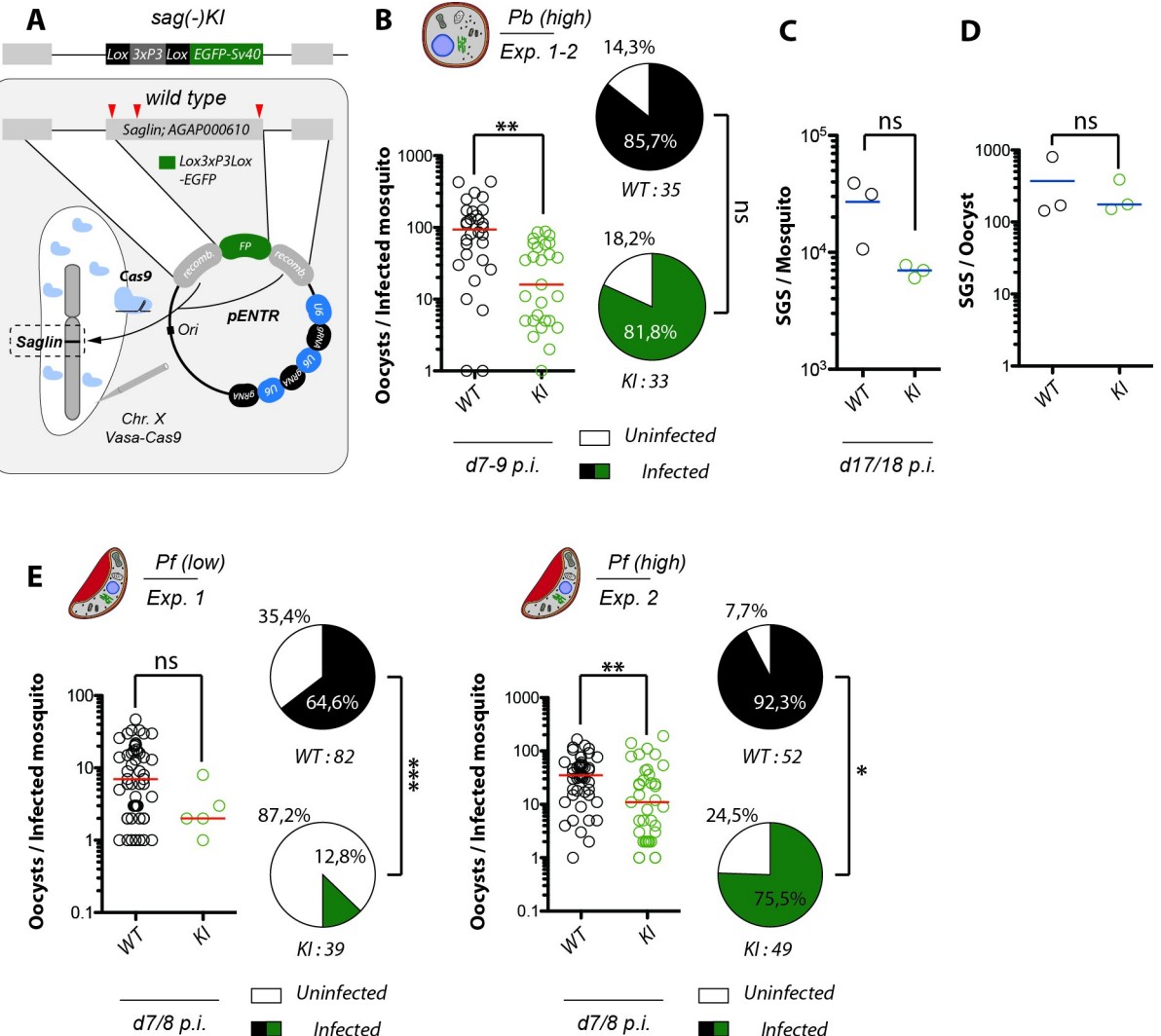

**Fig 2. Absence of Saglin impairs midgut colonization by *Plasmodium berghei* and *Plasmodium falciparum*. A)** Generation of *sag(-)KI* mosquitoes by Cas9 mediated site-directed mutagenesis inducing complete replacement of the *saglin* gene (AGAP000610) with a repair template encoding the fluorescence marker EGFP. Illustration not drawn to scale. Red arrowheads indicate binding sites of selected gRNAs. **B)** Oocyst densities in *sag(-)KI* and wild-type siblings infected with the rodent malaria parasite *P. berghei* (*Pb*). Pooled results of two independent experiments. The red line indicates the median. Mann Whitney test: \*\*p = 0.0013. Pie charts represent prevalences of infection. Fisher's exact test: not significant (ns). **C)** Means of the number of salivary gland sporozoites (SGS) per mosquito 17–18 days post infection from three independent infection experiments. The red line indicates the median. Paired t test: not significant (ns), p = 0.1290. **D)** SGS to oocyst ratio calculated based on paired experiments where both oocyst and salivary gland sporozoites were counted for both genotypes. The red line indicates the median. Paired t test: not significant (ns), p = 0.4431. **E)** Quantification of oocyst densities and prevalence of infection in *sag(-)KI* and control mosquitoes infected with *P. falciparum* in two different experiments. Data are presented as in B. The red line indicates the median. Mann Whitney test on oocyst densities. Exp. 1: not significant, p = 0.0751; Exp. 2: \*\*p = 0.0034. Fisher's exact test on prevalence of infection. Exp. 1: \*\*\*p < 0.0001; Exp. 2: \*p = 0.0286.

## Absence of Saglin expression has no impact on blood feeding behavior but impairs midgut colonization by *Plasmodium*

Functional depletion of saliva proteins can trigger blood feeding defects in *Anopheles* mosquitoes. For example, co-expression of a single-chain antibody against AAPP reduces the proportion of blood-feeding females as well as hemoglobin uptake, resulting in a lower number of eggs laid per female [40]. A decrease of the ingested blood volume would likely reduce the

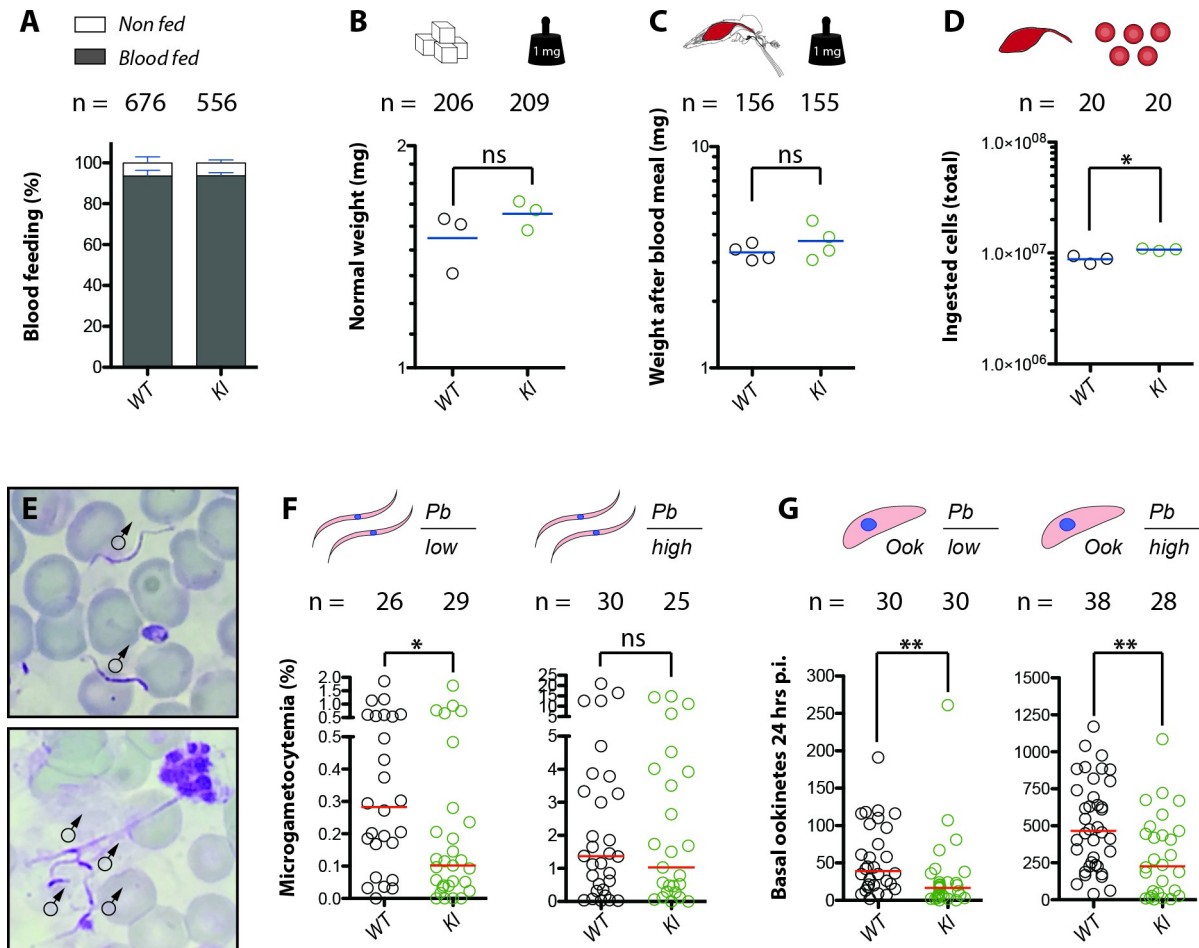

**Fig 3. Absence of Saglin impairs exflagellation and reduces the number of ookinetes successfully traversing the midgut. A)** Percentage of blood fed wild-type (WT) and *sag(-)KI* (KI) females after 10 minutes of feeding. Mean and standard error of the mean (SEM) of four independent experiments. **B)** Mean weight of females kept with sugar feeding (3 independent experiments). **C)** Weight of single females after blood feeding and **D)** red blood cell count (#RBC) in the blood meal after 10 minutes of blood feeding of wild-type (*Ngousso*) and *sag(-)KI* females. Data represent pooled results from four (blood feeding, #RBC) and three (sugar feeding) experiments, respectively. Blue lines indicate the mean. Experiments were performed ≥4 days after eclosion. Paired t test: ns = 0.1117 (weight sugar); ns = 0.3273 (weight blood); *p = 0.0158 (#RBC). Note that for all (A,B,C,D) but two of the four experiments of (C), larvae of *sag(-)KI* and wild type were reared as a mixture to avoid growth differences caused by unequal feeding. **E)** Diff-Quik stained smears of mosquito blood meals 10–12 minutes after feeding. Exflagellated male gametes are visible as elongated spirochete-like cells. Due to the recording method, it was not possible to draw a precise scale; red blood cells have an approximate diameter of 7 μm. **F)** Percentage of exflagellated male microgametes in the blood meal in relation to the total number of red blood cells (microgametocytemia) 10–12 minutes after feeding, and number of ookinetes (**G**) visible on the basal side of the midgut epithelium of wild-type (*Ngousso*) and *sag(-)KI* mosquitoes at 24 hpi (two independent experiments each). Mosquitoes were infected using mice with low (*Pb low*) or high (*Pb high*) parasitemia. Red lines indicate medians, with comparisons using the Mann Whitney test: ** p<0.01; * p<0.05; ns p>0.05. Note that the data in F and G have not been corrected for small differences in the ingested blood volume between wild-type and *sag(-)KI* mosquitoes.

number of uptaken gametocytes and lead to a reduced midgut colonization by *Plasmodium spp*. To test whether loss of Saglin affects blood feeding, homozygous *sag(-)KI* and wild-type females, bred either as a mixture or separately but always synchronously, were allowed to take blood on the arm of a human volunteer for ten minutes. Subsequently, blood-fed and non-fed females were counted for each genotype. Similar feeding rates of >90% were observed between *sag(-)KI* and wild-type females (**Fig 3A**). We next examined whether blood-fed *sag(-)KI* females ingested the same amount of blood as wild-type. For this, 206 wild-type and 209 *sag(-)KI* sugar fed females as well as 156 wild-type and 155 *sag(-)KI* blood-fed females were weighed.

Non blood-fed *Sag(-)KI* females were slightly, although not significantly, heavier than wild-type (**Fig 3B**). Accordingly, *sag(-)KI* females tended to have higher weight after blood feeding than wild-type females, although this difference was not significant. (**Fig 3C**). Unlike females of other mosquito species such as *Aedes spp.*, females of some *Anopheles* species raise the hematocrit of their blood meal by excreting fluid during blood ingestion [41]. Hence, blood meals of similar volumes may have different concentrations of red blood cells. Thus we also quantified the number of red blood cells in the blood meals of *sag(-)KI* and wild-type females. The cell count in blood meals of *sag(-)KI* females was significantly higher compared to wild-type females (**Fig 3D**), consistent with the observation that *sag(-)KI* females seem to uptake a greater volume of blood (**Fig 3C**).

We next investigated potential differences in the developmental progression of *P. berghei* in the mosquito midgut prior to the formation of oocysts. The only *Plasmodium* stages able to develop in the mosquito midgut upon blood feeding are female and male gametes. Activated male gametocytes divide in 8 microgametes that can easily be visualized using the same staining protocols as for asexual blood stages (**Fig 3E**). Here we determined the microgametocytemia in individual mosquito blood meals 10–12 minutes after ingestion. Mosquitoes were fed on highly (*Pb high*) and lowly (*Pb low*) parasitized mice. Quantification of blood meals obtained from *Pb low* infected mosquitoes revealed a significant decrease in the microgametocytemia of *sag(-)KI* compared to wild-type females (**Fig 3F**). Although the same tendency was observed in *Pb high* infected mosquitoes, the difference in the number of microgametes was not significant (**Fig 3F**). To determine whether differences may also be visible at the ookinete stage, the number of transmigrated ookinetes visible on the basal side of the midgut was determined ~24 h after blood feeding. In line with the decreased number of microgametes in *sag(-)KI* females, numbers of traversed ookinetes were significantly reduced in both *Pb low* and *Pb high* infected *sag(-)KI* mosquitoes (**Fig 3G**).

Taken together these results suggest no blood feeding reduction in *sag(-)KI* females, ruling out this possible explanation for reduced infection by *Plasmodium spp.*, and that Saglin rather promotes *Plasmodium* development in the mosquito midgut. We thus made use of the *sag(-)KI* reporter line to examine the Saglin expression pattern more precisely, notably in the midgut.

## Expression of Saglin is female-specific and restricted to the median lobes of the salivary glands

In *sag(-)KI* mosquitoes, the *EGFP* gene replacing the *saglin* coding sequence (**Fig 2A**), is placed under the control of the *3xP3* (driving expression in the eyes and nervous system) and *saglin* promoters, and functions as a reporter for transcriptional activity of the *saglin* promoter. The presence of both promoters upstream of *EGFP* thus leads to a *3xP3*-specific expression in the nervous system (in adults, mainly in the eyes) and a *saglin*-specific expression in the salivary glands. We created a second saglin knockout line called *sag(-)EX*, in which the *3xP3* promoter was removed by Cre-mediated excision, thus leaving *EGFP* under the control of the *saglin* promoter exclusively [19]. We could show that the *3xP3* promoter also affects the activity of the endogenous *saglin* promoter, by increasing reporter expression levels in the median lobe [19]. To investigate *saglin* promoter activity in salivary glands and midgut, we thus used the *sag(-)KI* line displaying the same expression pattern as *sag(-)EX* but giving stronger fluorescence signals, and the *sag(-)EX* when necessary to confirm the profile. In *sag(-)KI*, *EGFP* is strongly expressed in the salivary glands, to the point of being visible through the cuticle of the mosquito (**Fig 4A**). Promoter activity was observed in females but completely absent in males, indicating that Saglin is female-specific (**Fig 4A**) as previously reported for the mRNA

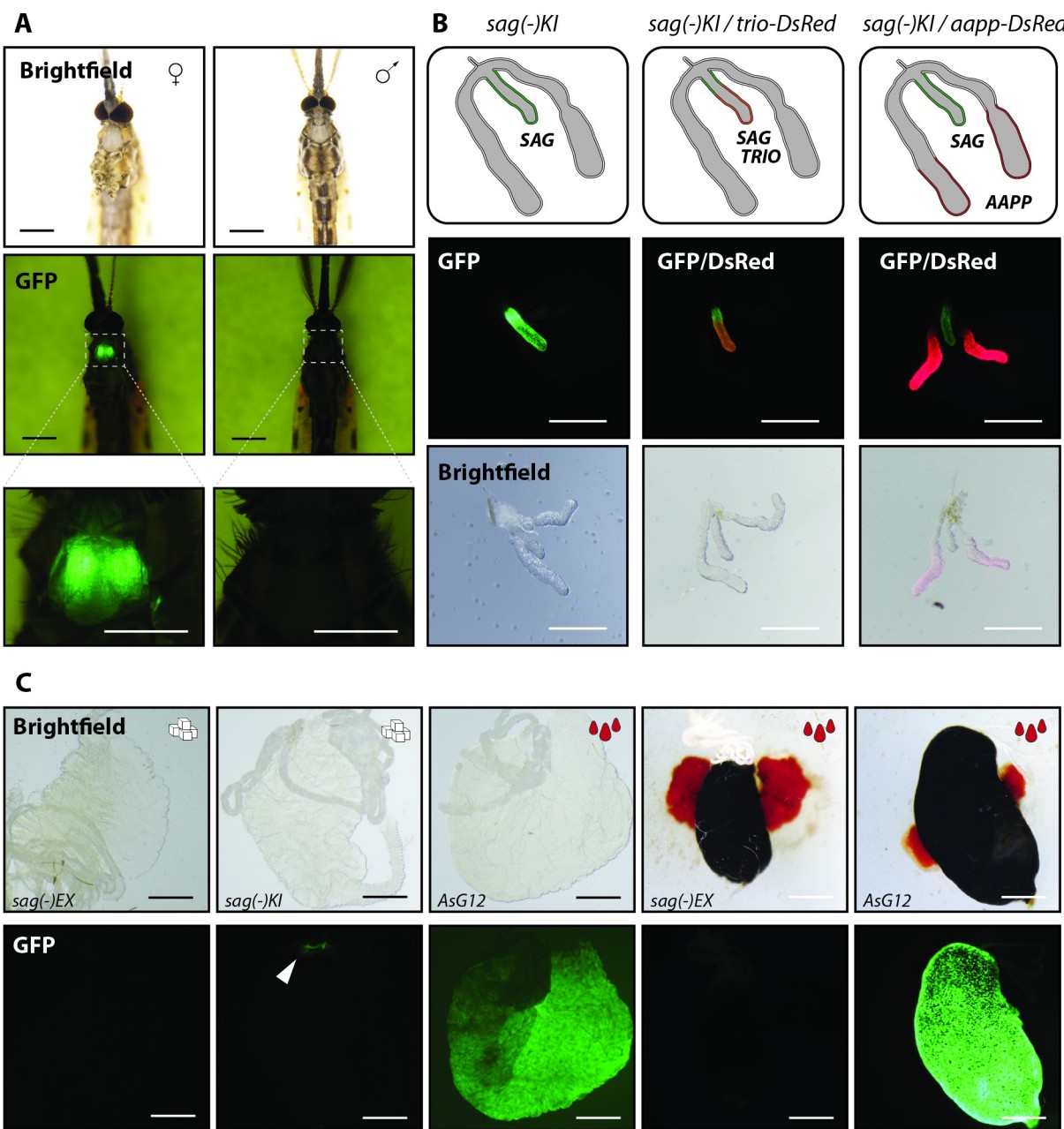

**Fig 4. Saglin is exclusively expressed in the median lobes of the salivary glands of female mosquitoes. A)** Brightfield images (top) and GFP signal in a homozygous *sag(-)KI* female and male mosquito. Scale bar: 0.5 mm. Bottom panels: close-ups of the GFP signal in the thorax. Scale bar: 250 μm. Male and female mosquitoes were imaged seven days after emergence using the same settings. **B)** *Sag(-)KI* females were crossed to males of the salivary gland reporter lines *trio-DsRed* and *aapp-DsRed*. Salivary glands of heterozygous F1 females were dissected and imaged. The genotypes as well as the expected expression patterns within the salivary glands are shown on the top. Fluorescence (middle) and brightfield (bottom) images of the same salivary glands. Scale bar: 250 μm. **C)** Comparison of EGFP expression in a sugar fed *sag(-)KI* midgut compared to sugar and blood fed *sag(-)EX* midguts. As positive control, midguts of the *An. stephensi* line *AsG12* which expresses EGFP under control of the blood meal inducible promoter G12 are shown [23]. Adults were dissected 7–10 days post emergence. Displayed *AsG12* midguts were dissected either four days after blood feeding (blood already digested) or 24hrs after blood feeding (blood filled). The white arrow indicates the EGFP expressing pyloric valve in *sag(-)KI* resulting from activity of the 3xP3 promoter. Scale bar: 250 μm.

transcript [18]. Dissections of salivary glands obtained from *sag(-)KI* females revealed that the *saglin* promoter is only active in the median lobe of each salivary gland, with no *EGFP* expression in the proximal- and distal-lateral lobes (**Fig 4B**). This very specific expression pattern became evident by crossing *sag(-)KI* mosquitoes with the salivary gland reporter lines *trio-DsRed* and *aapp-DsRed* expressing *DsRed* exclusively in the median and distal-lateral lobes, respectively [19]. Accordingly, the EGFP signal colocalized with DsRed exclusively in the salivary glands obtained from females heterozygous for *sag(-)KI* and *trio-DsRed*, but not *aapp-DsRed* (**Fig 4B**). As blood feeding can specifically induce expression of genes important for blood digestion in the midgut [23] where Saglin could potentially directly interact with *Plasmodium* parasites ingested by blood feeding, we assessed *EGFP* expression in the midgut 24h after blood feeding using the *sag(-)EX* line (**Fig 4C**). Indeed, we had noticed EGFP fluorescence in cells of the pyloric valve of sugar fed *sag(-)KI* midguts. Other transgenic mosquito lines expressing fluorescent reporters under control of the *3xP3* promoter also present this fluorescence pattern, suggesting that it is *3xP3*-specific. No EGFP signal was detected in blood-fed and sugar-fed *sag(-)EX* midguts, while blood-fed midguts of the *An. stephensi* line *As-G12* expressing GFP under control of the blood meal inducible *G12* promoter showed strong and uniform GFP expression (**Fig 4C**) [23]. Taken together, our results indicate that Saglin is specifically expressed in the salivary gland median lobes of female mosquitoes.

## Saglin is expressed in the salivary glands, excreted during blood feeding and re-ingested with the blood meal

A previous study reported that Saglin might function as a receptor mediating recognition and invasion of the salivary glands by *Plasmodium* sporozoites [9]. Because sporozoites preferentially invade the lateral lobes [42], we investigated whether Saglin can spread into the lateral lobes of the salivary gland, and whether it could be exposed on the salivary gland surface to contact sporozoites by performing immunofluorescence experiments with α-Saglin antibodies. Similar to the *EGFP* expression in *sag(-)KI* mosquitoes, Saglin-specific staining of wild-type salivary glands was observed in the acinar cells of the median lobe (**Fig 5A**). Of note, due to the strong polarity of the acinar cells that form an inward cavity towards the salivary duct, their cytoplasm is restricted to the edges of the median lobe. Accordingly, Saglin-specific signals were observed in the marginal area, likely present in the acinar cell cytoplasm although we cannot exclude that a subset of Saglin is localized on the surface of the median lobe. In addition, we observed very strong Saglin signals in the lumen of the proximal region of lateral lobes (**Fig 5A**), although cells in these regions did not show any *saglin* promoter activity (**Fig 4B**). This observation is in line with previous findings showing a similar immunofluorescence staining pattern for Saglin in *An. stephensi* [17]. Interestingly, fluorescence signals in the proximal-lateral lobes were stronger compared to the median lobe, which could indicate differences in protein concentrations, or in the accessibility of the α-Saglin antibodies when the protein is intracellular or secreted. As expected, salivary glands obtained from homozygous *sag(-)KI* females displayed no Saglin signal (**Fig 5A**). We further hypothesized that in WT mosquitoes, Saglin might be secreted into the hemolymph and acting as a chemoattractant for sporozoites to find the salivary glands, or that the median lobe might serve as an entry point from which sporozoites translocate to the lateral lobes. To assess this, we injected XFD594-labeled wheat germ agglutinin (WGA) into live mosquitoes to stain the surface of the salivary glands that is in contact with hemolymph, which would likely be a requirement for saglin secretion in the hemolymph and for parasite invasion. Imaging 13 dissected salivary glands revealed a uniform staining of distal-lateral and proximal-lateral lobes while >50% of the median lobes were devoid of WGA coating (**Figs 5B and S3**). When median lobes displayed fluorescence, the

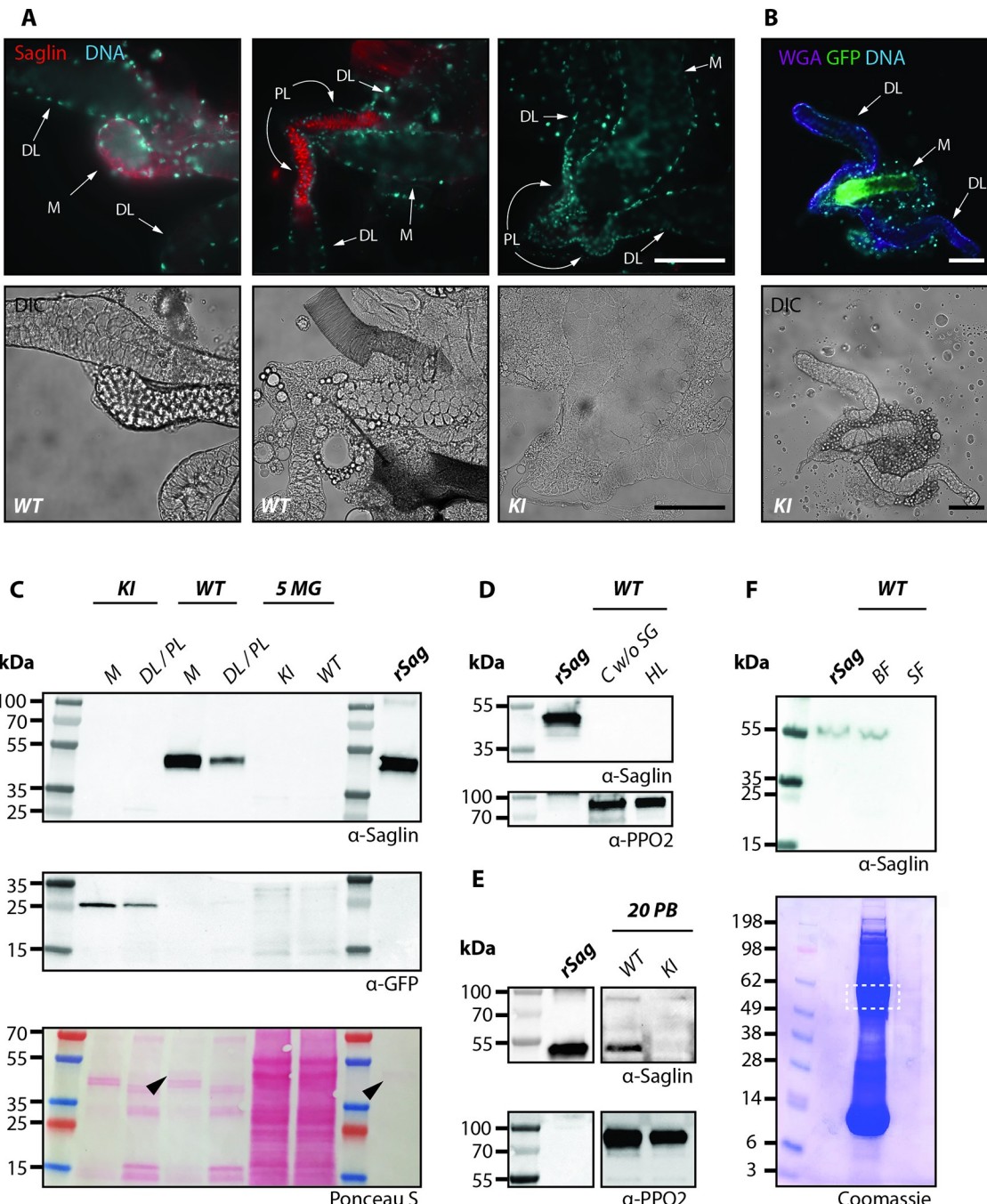

**Fig 5. Saglin protein is found in the salivary glands, excreted with saliva and re-ingested with the blood meal. A)** Immunofluorescence staining of Saglin in fixed wild-type (*Ngousso*) and *sag(-)KI* salivary glands. Saglin (red) and DNA (cyan) stainings (top), and differential interference contrast (DIC, bottom). Images depict the same salivary glands. DL: distal-lateral lobes; PL: proximal-lateral lobes; M: median lobe. Scale bar: 100 μm. WT: wild-type. KI: *sag(-)KI*. **B)** Staining with fluorophore coupled wheat germ hemaglutinin (WGA) of a *sag(-)KI* salivary gland *ex vivo*. WGA staining in purple, DNA in cyan and *EGFP* expression under the control of the *saglin* promoter in green. Scale bar: 100 μm. **C-F)** Western blots of different mosquito tissues to detect the presence of Saglin. **C)** Median (M) and distal-lateral/proximal-lateral lobes (DL/PL) in comparison to unfed midguts (MG) of the same wild-type (*WT*) and *sag(-)KI* (KI) mosquitoes. Arrowheads indicate Saglin size in wild-type median lobes and recombinant Saglin lanes on the Ponceau S stained blot. **D)** Carcasses with salivary glands removed and hemolymph of wild-type. **E)** Proboscis of wild-type and *sag(-)KI* (PB). Two different images of the same blot to account for removed lanes. **F)** Bolus of sugar (SF) and blood fed (BF) wild-type midguts. The dashed square indicates the area where Saglin is expected to localize in the Coomassie stain. For all Western blots, recombinant Saglin (rSag) was used as positive control. Western blots and SDS-Page gels were treated with α-PPO2 antibodies or Ponceau S, and Coomassie blue, respectively, to assess loading of samples.

signal was either weak compared to lateral lobes, or restricted to the tip (**S3 Fig**). These results suggest that the Saglin-expressing median lobe is less accessible than the lateral lobes from the hemolymph. Of note, and in contrast to lateral lobes, the median lobe was often tightly embedded in fat body tissues, which may in part explain its lower accessibility. WGA staining was specific upon injection and no difference in the staining pattern was observed between wild-type and *sag(-)KI* mosquitoes (**S4 Fig**). To further confirm the salivary gland-specific activity of the *saglin* promoter and fate of Saglin, we performed a series of western blot experiments on dissected tissues (**Fig 5C–5F**). Staining with α-Saglin antibodies revealed a strong band for median lobes and a faint band for lateral lobes obtained from wild-type mosquitoes. No Saglin was detectable in salivary gland samples obtained from *sag(-)KI* mosquitoes (**Fig 5C**), confirming that the mutant is a full knockout. Similarly, EGFP was highly enriched in the median lobe sample of *sag(-)KI* mosquitoes with a fainter signal in lateral lobes (**Fig 5C**). No Saglin was detected in midguts (**Fig 5C**), hemolymph and carcasses deprived of salivary glands of sugar fed wild-type females (**Fig 5D**), likely excluding any *saglin* expression in other tissues as well as Saglin dissemination from the salivary glands. Saglin was detected in cut proboscises collected from wild-type but not *sag(-)KI* females (**Fig 5E**), suggesting that Saglin is a component of saliva and potentially excreted during salivation. In line with these observations, a previous study using mass spectrometry detected Saglin in the saliva of mosquitoes [14]. Thus, we hypothesized that Saglin might be re-ingested during blood intake. Indeed, we could detect Saglin in the midgut bolus of blood-fed wild-type but not in sugar-fed wild-type and blood fed *sag(-)KI* females, revealing that Saglin is present in the gut, but only after blood feeding (**Fig 5F**). Taken together, our data suggest that Saglin is specifically expressed in the median lobe of the salivary glands, probably stored in the duct of the proximal-lateral lobes and injected in the host skin upon salivation, where it can be re-ingested by the mosquito during blood intake. Therefore, we hypothesized that Saglin ingested with the blood meal may have a pro-parasitic effect on *Plasmodium* midgut colonization. If so, the observed *Plasmodium* infection reduction in *sag(-)KI* mosquitoes may be rescued by the presence of recombinant Saglin (rSag) or saliva from wild-type mosquitoes.

As previously observed, we confirmed that the *sag(-)KI* phenotype is visible when a mixture of wild-type and *sag(-)KI* mosquitoes is fed together on the same mouse (**S5 Fig**). This was also the case for experiments where mice were first bitten by wild-type mosquitoes before infection of both genotypes on the same mouse (**S5 Fig**), suggesting that the quantity or availability of Saglin salivated by wild-type mosquitoes is not sufficient to compensate for the lack of Saglin in *sag(-)KI* mosquitoes feeding next to, or right after, a wild type mosquito. We next injected rSag intravenously in parasitized mice about 10 min before infecting mosquitoes (**S5 Fig**). However again, both prevalence and intensity of infections in *sag(-)KI* females remained lower compared to wild-type. Several technical issues may explain this absence of rescue: (1) rSag may not be functional, (2) the timing between rSag injection and blood feeding may not be optimal, or (3) the concentration of injected protein may not be high enough. Of note, rSag appears slightly lighter than salivary gland Saglin on Western blots, indicating that rSag, produced in bacteria, may be devoid of potential posttranslational modifications important for its function [14]. To circumvent this issue, we assessed whether the perfusion of salivary gland extracts from wild-type mosquitoes could complement the *sag(-)KI* phenotype (**S5 Fig**). As above, we could not detect a rescue effect in the *sag(-)KI* phenotype.

## Saglin promotes successful transmission at low infection densities

To test whether the decrease in parasite load observed *in sag(-)KI* mosquitoes could affect mosquito capability to transmit *Plasmodium* sporozoites, we performed mouse infection

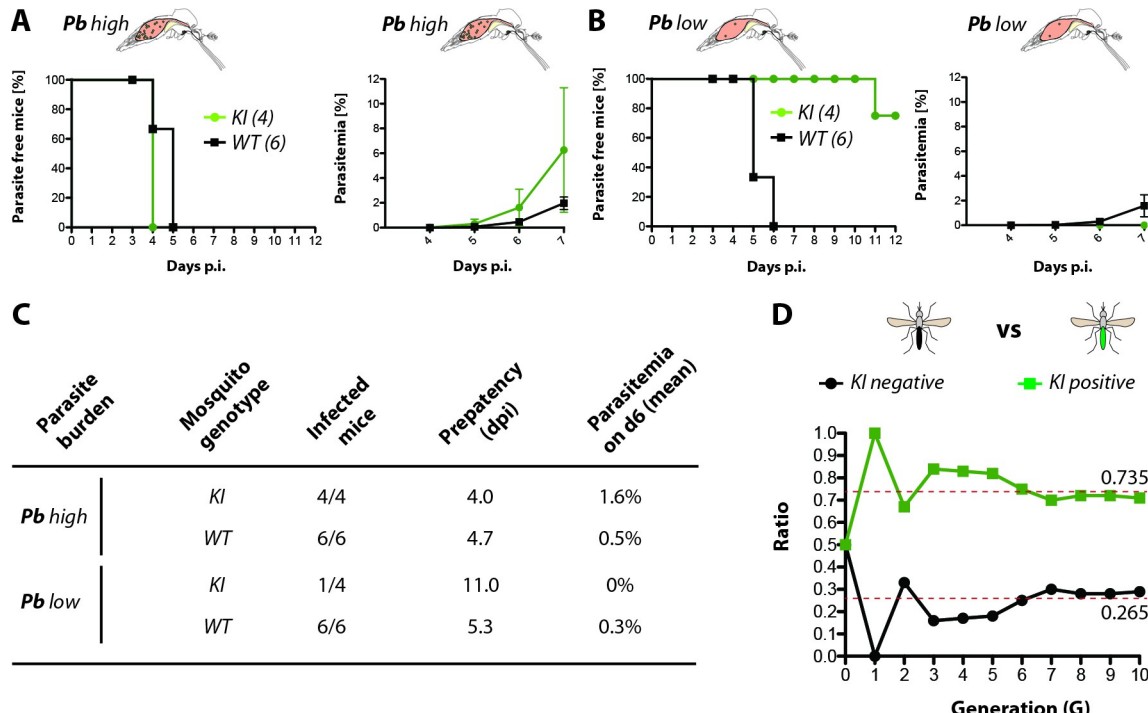

**Fig 6. Saglin promotes successful transmission at low infection densities. A)** Percentage of parasite free mice after being exposed to the bites of ten *sag(-)KI* or wild-type mosquitoes with high (*Pb* high) or **B)** low (*Pb* low) parasitemia. The number of mice is given in brackets. The growth of asexual blood stages in infected mice was monitored between day 4 and day 7 after mosquito exposure. **C)** Table summarizing results illustrated in A and B. Mosquito genotypes, numbers of infected / exposed mice as well as mean prepatency in days and mean parasitemia at day 6 post infection are indicated. Prepatency is defined as the time between the mosquito bite and the first observation of parasite blood stages. **D)** Evolution of the presence of the *sag(-)KI* transgene in a colony established by crossing homozygous *sag(-)KI* females to wild-type males (G0) over 10 generations. The proportion of transgenic (homozygote + heterozygous females and hemizygous males) and wild-type larvae was tracked by flow cytometry (COPAS) at each generation, using the 3xP3-GFP fluorescence marker disrupting the *saglin* coding sequence. Dotted red lines indicate the expected ratios for an X-linked allele in Hardy-Weinberg equilibrium.

experiments using wild-type and *sag(-)KI* mosquitoes infected with *P. berghei*. Because *P. berghei* infections in *Anopheles spp.* mosquitoes generally cause a much higher parasite load compared to *P. falciparum*, two different infection regimes were chosen to achieve high and low parasite loads while maintaining high prevalence. Highly infected mosquitoes (*Pb high*) reflect common *P. berghei* infection levels achieved under laboratory conditions, while mosquitoes with a low parasite load (*Pb low*) reflect parasite numbers more commonly seen in mosquitoes infected with *P. falciparum* in the wild. For this, mosquitoes were infected on mice showing a parasitemia of either <1% (*Pb low*) or >2% (*Pb high*). *Pb high*-infected wild-type and *sag(-)KI* mosquitoes displayed a transmission efficiency of 100%. Of the 6 mice bitten by *Pb high*-infected wild-type mosquitoes, all became positive after a prepatent period of 4.7 days. Similarly, 4 of 4 mice bitten by *Pb high*-infected *sag(-)KI* mosquitoes became positive after a prepatent period of 4 days (**Fig 6A and 6C**). Consistent with the observed difference of 0.7 days in prepatency, mice infected by *sag(-)KI* mosquitoes showed faster parasite growth and, on average, higher parasitemia at day 6 post infection compared with mice infected by wild-type mosquitoes (**Fig 6A and 6C**). In contrast, while 6 of 6 mice bitten by *Pb low*-infected wild-type mosquitoes became positive for blood stage parasites after a prepatent period of 5.3 days, only 1 of 4 mice bitten by *Pb low*-infected *sag(-)KI* mosquitoes became parasite positive after 11 days indicating a severe impairment in the transmission capacity (**Fig 6B and 6C**). Finally, to

assess the fitness of *sag(-)KI* mosquitoes, which is a crucial parameter for transmission ability under natural conditions, we examined the persistence of the *sag(-)KI* transgene in competition with the wild-type locus. For this, we crossed females homozygous for the *sag(-)KI* transgene with wild-type males. The ratio of individuals positive and negative for the *sag(-)KI* transgene was tracked by flow cytometry over 10 generations, revealing that the frequency of transgene carriers and non-carriers did not deviate from the Hardy-Weinberg equilibrium for X-linked inheritance over this time (**Fig 6D**).

## Discussion

The knockdown of *saglin* has been shown to lower sporozoite burdens in the salivary glands of *P. falciparum* infected mosquitoes, which has been attributed to an impaired interaction with the sporozoite-specific protein TRAP [9]. Here we confirm that the complete loss of Saglin in transgenic mosquitoes decreases sporozoite loads in the salivary glands. However, our data rather support a role of Saglin during parasite development in the midgut, with limited to no contribution to salivary gland invasion by sporozoites. We provide several lines of evidence indicating that Saglin facilitates parasite invasion of the mosquito midgut. *Sag(-)KI* mosquitoes display a reduced oocyst burden compared to wild-type, which may be in part explained by a reduction in male gametocyte exflagellation and in the number of ookinetes able to cross the midgut epithelium. In contrast, we could not detect a specific effect of Saglin knockout on salivary gland invasion by sporozoites. Indeed, the decreased sporozoite burden in *sag(-)KI* mosquitoes was proportional to the decreased oocyst burden. Interestingly, a similar phenotype was recently described for the knockout of the salivary protein SGS1 in *A. aegypti*. Knockout mosquitoes lacking SGS1 expression showed a 48–79% decrease in oocyst numbers upon infection with *P. gallinaceum* [11]. This observation is puzzling since feeding or intrathoracic injection of antibodies directed against Saglin, or injection of the SM1 peptide that interacts with Saglin, reduce the capacity of sporozoites to invade the salivary glands [9,16]. Hence the fact that *sag(-)KI* mosquitoes show no impairment in the passage of sporozoites from the midgut to the salivary glands suggests that the antibody and SM1 may impact salivary gland invasion by a different mechanism than a physical blockade. Supporting this hypothesis, antibodies against CSP-BP have the same effect as antibodies against Saglin, although CSP-BP is believed to have an exclusively intracellular function [8]. Immunofluorescence stainings provided here as well as in a previous study performed in *An. stephensi* [17] demonstrate that Saglin does not localize at the salivary gland distal-lateral lobes where the majority of sporozoites invade [42]. Saglin could be detected at the edges of the median lobe by immunofluorescence, raising the possibility that a subset of Saglin is exposed at the surface. However, injection of fluorophore-labelled WGA in living mosquitoes mainly stained the salivary gland lateral lobes, indicating that the median lobe has little contact with hemolymph. In addition, the median lobes are usually embedded in fat tissue that is difficult to remove during dissection. Taken together, these observations could, at least in part, explain why the lateral lobes are preferentially infested by sporozoites [42]. If this is true, the limited physical access to the median lobe would further limit the ability of surface-exposed Saglin to function as a sporozoite receptor. Furthermore, Saglin is not secreted at detectable levels in the hemolymph where it could potentially serve as a chemotactic attractant for sporozoites. In addition, a recent study has shown that sporozoites do invade the salivary glands of male *Anopheles* mosquitoes [43] although *saglin* expression is female-specific [18,19]. Taken together, these observations indicate that Saglin is unlikely to be a determinant for sporozoite invasion of the salivary glands as sporozoites should not come in contact with this protein before reaching the lumen of salivary glands. There, TRAP may still have a capacity to interact with Saglin. Indeed, TRAP has been

shown to bind ubiquitously to a wide range of different proteins such as fetuin-A on hepatocytes [44], alpha-v-containing integrins [45] and platelet derived growth factor receptor β [46]. Therefore, it is conceivable that a Saglin-TRAP interaction in the skin increases sporozoite transmissibility, independently from the Saglin effect on the gametocyte/ookinete stages. This remains to be investigated.

The comparison of the expression pattern of the *EGFP* reporter under control of the *saglin* promoter in the *sag(-)KI* line with immunofluorescence images revealed that Saglin accumulates at high concentrations in the lumen of the proximal region of the lateral lobes while *saglin* promoter activity is restricted to the median lobe. This and the fact that Saglin has a signal peptide, suggest that Saglin is secreted into the salivary duct, where it spreads to the proximal lobes, likely to be stored and mixed with other saliva components in preparation for salivation. The notion that Saglin is a component of saliva is supported by mass spectrometric detection of Saglin in saliva samples [14]. We did not detect Saglin expression (*EGFP* expression in *sag (-)KI* and *sag(-)EX* mosquitoes) or protein (by Western blotting) in any other tissues of female mosquitoes (hemolymph, midgut, carcasses deprived of salivary glands), including after blood feeding, demonstrating that Saglin is specifically expressed in the salivary glands and confined there. Hence the question, how can Saglin affect parasite development in the midgut if it is solely expressed and resident in the salivary glands? The presence of Saglin in the proboscis and in the gut of wild-type mosquitoes just after blood intake suggests that Saglin is indeed injected to the host skin at the bite site together with other saliva compounds, and re-ingested together with blood. We thus hypothesize that Saglin may interact with gametocytes at the bite site or in the ingested blood meal. For example, it has been demonstrated that *Ae. aegypti* mosquitoes re-ingest much of their own salivary secretions [47]. This observation is in line with the finding that a significant number of salivary gland sporozoites can be found in the midguts of *An. stephensi* and *An. gambiae* mosquitoes after blood feeding [48]. Still, our priming experiments to allow *sag(-)KI* mosquitoes to uptake traces of Saglin left in the skin by wild-type females, did not rescue *Plasmodium* development to wild-type levels, possibly because bite sites are too distantly positioned and Saglin not available in high enough quantities to complement Saglin deficiency in *sag(-)KI* mosquitoes. Similarly, the perfusion of parasitized mice with salivary gland extracts obtained from wild-type females or with recombinant Saglin prior to infection did not rescue the *sag(-)KI* phenotype. Although these approaches seemed more promising, interpretation of the data is difficult since rSag was expressed in *E. coli* and is lacking posttranslational modifications that may be important for its function, and that its functionality cannot be assessed. Indeed, Saglin has been predicted to be glycosylated at N95 [14]. Similarly, the stability and concentration of injected endogenous Saglin in mouse blood are unknown after perfusion of salivary gland extracts. The timing between perfusion and infection, and the amount of salivary gland extracts may be non-optimal to achieve an effect at the bite site. Rescue experiments might be improved by performing infection experiments using artificial feeders and higher amounts of salivary gland extracts, or rSag produced in a eukaryotic expression system.

If Saglin is important for parasite development in the mosquito midgut, two different modes of action are conceivable. First, Saglin may function as a mosquito-specific cue for gametocyte activation and trigger life cycle progression similarly to xanthurenic acid [49]. In this scenario, the intrinsic function of Saglin would be uncoupled from its effect on the parasite. Second, Saglin could alter physiological conditions in the blood meal, stimulating *Plasmodium* development. In this case the proparasitic effect of Saglin may not require a direct parasite/Saglin interaction. No function has yet been assigned to Saglin or any other protein in the SG1 family. Still, we can already draw some conclusions from our data. The knockout of Saglin has no negative effect on blood feeding or on the fitness of mosquitoes under laboratory

conditions. In contrast, the suppression of salivary proteins involved in the inhibition of blood coagulation and platelet aggregation, such as the anopheline antiplatelet protein (AAPP) and Aegyptin, impair blood feeding behavior and fecundity [40,50]. Interestingly, we also observed that reduced infection in *saglin*-knockout mosquitoes applies to both the rodent malaria parasite *P. berghei* and the human malaria parasite *P. falciparum*. While mosquitoes were allowed to feed directly on mice for *P. berghei* infections, *P. falciparum* infections were performed using gametocyte cultures and artificial feeding devices. Since *P. falciparum* cultures lack cellular components of the immune system and contain only inactivated complement factors, the possibility that Saglin may be involved in the neutralization of immune factors, thereby protecting the parasites, is less likely. No structure of a member of this protein family has been solved, however *in silico* structural prediction of monomeric Saglin using I-Tasser [51] revealed that the nucleotide binding domain 1 linked to the middle domain of monomeric Hsp104 from *S. cerevisiae* (PDB: 6AHF) is the closest related structure followed by other chaperones like *Ct*Hsp104 (PDB: 5D4W) and *Tt*ClpB (PDB: 1QVR). Interestingly, the salivary D7 protein from *Culex quinquefasciatus* was shown to bind the nucleotides adenosine diphosphate (ADP) and adenosine triphosphate (ATP) with high affinity. Since ADP and ATP play an important role in activating platelet aggregation, their depletion in the blood meal by binding to D7 enhances blood feeding on mammals [39]. Other D7 proteins sequester hemostasis mediators such as serotonin and epinephrine with different affinities based on very different ligand specificities [52]. SG1 proteins share limited sequence similarity with each other [18] but contain several highly conserved residues like C18 and C60 (positions referring to Saglin, AGAP000610) and amino acid stretches and share a similar size (385–431 aa) (**Figs 1B and S1**). Conserved cysteines together with a similar size might indicate that secondary and tertiary structural elements are at least partially conserved. This could indicate a conserved mode of action of the SG1 family, possibly similar to the D7 proteins that sequester very different molecules, although the two protein families do not share significant sequence similarity.

The absence of a fitness cost in Saglin loss-of-function mutants in combination with a decrease of 75% in transmission capacity for *P. berghei* makes Saglin an attractive target for gene drive approaches. Disruption of the *saglin* gene with a driving cassette is expected to decrease the malaria transmission ability of mosquitoes without the need for additional anti-*Plasmodium* effectors. Still, *P. berghei* parasites that managed to invade *sag(-)KI* salivary glands were infectious to mice, thus an additional effector could be introduced to further reduce the transmission capacity of *sag(-)* mosquitoes. We do not know how Saglin affects parasite development in the mosquito midgut, but we showed that Saglin absence reduces the number of microgametes and ookinetes invading the midgut epithelium. Further characterization of Saglin and other SG1 proteins will be instrumental to refine transmission blocking strategies. Should it be feasible to uncouple the intrinsic function of Saglin from its effect on parasites, the construction of transmission incompetent Saglin alleles retaining their intrinsic function may become possible, thereby eliminating the need for gene knockout and reducing the potential impact of genetic interventions on mosquito fitness in the field.

## Supporting information

**S1 Fig. Multiple sequence alignment of all seven SG1 proteins known in *An. gambiae*.** The initial alignment was performed with PSI-Coffee [55] while sequence similarities based on physico-chemical properties of amino acids were calculated with ESPript [56]. The colored stripe above the alignment indicates the quality of the alignment according to ESPrit. In addition the secondary structure of Saglin (AGAP000610) predicted by I-Tasser is shown along the alignment [51]. Please note that the signal peptides from all proteins except AGAP000611, for

which no signal peptide has been predicted, were removed in preparation of the alignment.
(TIF)

**S2 Fig. The *sag(-)KI* phenotype is retained even if knockout and wild-type colonies are maintained separately.** Oocyst densities in *sag(-)KI* and wild-type (*Ngousso*) mosquitoes derived from two different colonies. Results of two pooled experiments. The red lines indicate medians, comparison using Mann Whitney test: ***p = 0.0007. Pie charts represent prevalence of infection. Fisher's exact test: *p = 0.0394.
(TIF)

**S3 Fig. Quantification of *in vivo* WGA staining patterns of median and lateral lobes. A)** Staining pattern of SGS in mosquitoes injected with wheat germ agglutinin (WGA) coupled to XFD594. WGA-XFDS594 was injected into living mosquitoes using a capillary and mosquitoes were dissected one hour after injection. Lateral lobes (both distal DL and proximal, PL) were stained in all samples. Staining patterns in median lobes (M) were classified as „apex", „weak"- and „unstained". Images illustrating each pattern are given showing localisation of WGA-XFD594 (top) and differential interference contrast (DIC, bottom). Scale bar: 100 μm. **B)** Quantification of staining patterns according to (A). Thirteen salivary glands were analyzed.
(TIF)

**S4 Fig. WGA coating is not affected by the absence of Saglin.** Staining patterns of salivary glands dissected from female mosquitoes injected with wheat germ agglutinin (WGA) coupled to XFD594. The salivary gland staining of an injected female homozygous for *sag(-)KI* is compared to two salivary glands dissected from an injected and a non-injected wild-type (*control*) female. Scale bar: 100 μm. Columns from left to right: mosquitoes genotypes, differential interference contrast (DIC), WGA and GFP signal in black on white; merge of WGA (red), GFP (green) and nucleic acid staining (DNA). Red arrowheads indicate median lobes.
(TIF)

**S5 Fig. Priming of the bite site and intravenous injection of recombinant Saglin and salivary gland extracts do not rescue infection levels in *sag(-)KI* mosquitoes. A)** Co-feeding of *sag(-)KI* (*KI*) and wild-type (*WT*) mosquitoes on the same *P. berghei* infected mouse, infections of *sag(-)KI* and wild-type mosquitoes on mice "primed" with wild-type and *sag(-)KI* and intravenous (i.v.) injection of PBS, recombinant Saglin (rSag) and salivary gland extracts. **B)** Each dot represents one experiment, 1–4 independent experiments for each group. Solid blue lines indicate the means, and the dashed blue line a ratio of 1 (expected if the *sag(-)KI* phenotype is rescued). **C)** Prevalence of infection for experiments shown in B. Shown is the mean with the standard error of the mean (SEM) in blue.
(TIF)

## Acknowledgments

We thank Ludivine Ramolu for technical support, Sarra Manai for help with western blot experiments, Lionel Brice Feufack Donfack for help with *P. falciparum* infections and Dr. Jean-Daniel Fauny for assistance during microscopy. We also thank Dr. Paola Valenzuela-Leon and Karina Botello for assistance with midgut Western blots. We would like to thank the mosquito immune responses (MIR) team for fruitful discussions and for assistance with mosquito breeding. We also thank the CNRS, Inserm and the University of Strasbourg for providing the infrastructure, salaries and for their support.

## Author Contributions

**Conceptualization:** Dennis Klug.

**Data curation:** Dennis Klug, Amandine Gautier.

**Formal analysis:** Dennis Klug, Amandine Gautier.

**Funding acquisition:** Dennis Klug, Eric Calvo, Eric Marois, Stéphanie A. Blandin.

**Investigation:** Dennis Klug.

**Methodology:** Eric Calvo, Eric Marois.

**Project administration:** Dennis Klug.

**Resources:** Amandine Gautier, Eric Calvo, Eric Marois, Stéphanie A. Blandin.

**Supervision:** Dennis Klug, Stéphanie A. Blandin.

**Validation:** Dennis Klug.

**Visualization:** Dennis Klug.

**Writing – original draft:** Dennis Klug.

**Writing – review & editing:** Eric Calvo, Eric Marois, Stéphanie A. Blandin.

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
