## [Decision Letter · Decision Letter 0]

7 Jul 2022

Dear Mr. Klug,

Thank you very much for submitting your manuscript "The salivary protein Saglin facilitates efficient midgut colonization of Anopheles mosquitoes by malaria parasites" for consideration at PLOS Pathogens. As with all papers reviewed by the journal, your manuscript was reviewed by members of the editorial board and by several independent reviewers. In light of the reviews (below this email), we would like to invite the resubmission of a significantly-revised version that takes into account the reviewers' comments.

Reviewers 1 and 2 both criticize the lack of direct evidence that ingested saglin affects parasite development in the midgut. Addressing this point with additional data would help support the most novel aspect of your work and increase its impact significantly. The obvious experiment would be to ask if your recombinant saglin, if added to an infectious feed or otherwise applied to the gut, complements the saglin knockout. We would consider such direct evidence important to support your conclusion that saglin must be re-ingested to affect the parasite. In the absence of additional evidence, or if the data do not support your proposed mechanism, please phrase your statement in the abstract and elsewhere more carefully. 

Reviewer 1 also raises the question of the mechanism by which saglin affects oocyst numbers, which you cover in your discussion. Further insights would increase the impact of the manuscript, but we would consider this outside the scope of the current analysis. 

Please pay special attention to the summary section of Reviewer 3. While this reviewer gets the expected mass of mosquitoes wrong, they raise important methodological concerns that we ask you to consider and address carefully. Uncertainties and variance in measurements should be noted and propagated through calculations where appropriate. 

Fig. 3B is said to show “Numbers of exflagellated male microgametes” but the axis label reads “microgametocytaemia (%)”. What is it, and if it is the former, what is the unit?

We cannot make any decision about publication until we have seen the revised manuscript and your response to the reviewers' comments. Your revised manuscript is also likely to be sent to reviewers for further evaluation.

Sincerely,

Oliver Billker

Associate Editor

PLOS Pathogens

Kirk Deitsch

Section Editor

PLOS Pathogens

Kasturi Haldar

Editor-in-Chief

PLOS Pathogens

orcid.org/0000-0001-5065-158X

Michael Malim

Editor-in-Chief

PLOS Pathogens

orcid.org/0000-0002-7699-2064

Reviewer's Responses to Questions

**Part I - Summary**

Reviewer #1: In this study, Klug et al revisit the role of Saglin during Plasmodium infection in mosquitoes. Saglin has been previously identified as a key protein required for colonization of A. gambiae salivary glands by P. falciparum sporozoites, acting as a receptor for the parasite protein TRAP (Ghosh et al Plos Pathogens 2009). However, a more recent study showed that saglin is expressed mainly in salivary gland median (but not lateral) lobes and that induction of saglin expression in lateral lobes does not influence P. falciparum infection in transgenic A. stephensi mosquitoes, questioning the role of saglin (O’Brochta et al Malaria Journal 2019).

Here the authors generated saglin knockout A. gambiae mosquitoes. The absence of saglin caused a decrease in P. berghei and P. falciparum oocyst numbers, but no specific effect on salivary gland colonization by sporozoites. The authors found that knockout of saglin has no effect on mosquito blood feeding, but is associated with reduced numbers of male gametes and traversing ookinetes in the midgut. This translates into reduced parasite transmission at low infection densities in the P. berghei-mouse model. The authors analyzed saglin expression pattern, based on promoter activity and immunofluorescence assays, revealing that the protein is female-specific and restricted to the median lobes of the salivary glands, consistent with previous observations (O’Brochta et al Malaria Journal 2019). They also report that saglin is excreted in the saliva and re-ingested with the blood meal during blood feeding. Based on these observations, they propose that saglin ingested with the blood meal may promote infection in the midgut. Finally, the data show that saglin knockout has no fitness cost in mosquitoes, indicating that this gene could be a potential candidate for gene drive strategies.

The manuscript is well written and the work is very well executed. The data provide compelling genetic evidence ruling out a role for saglin during colonization of the salivary gland by Plasmodium sporozoites. Interestingly, the data also indicate that saglin plays a role at an earlier step, during infection of the mosquito midgut. The authors propose that saglin present in the saliva is re-ingested during the blood meal and may promote establishment of infection in the midgut. This is an interesting model and the main novelty here. However, this part of the study would benefit from a deeper investigation of saglin contribution during midgut infection. In addition, the manuscript would be considerably strengthened if the authors could provide direct evidence that ingested saglin promotes midgut infection.

Reviewer #2: The work by Klug and colleagues investigates the importance of the Anopheles protein saglin for the development of Plasmodium inside its mosquito vector. Saglin supposedly acts as a receptor for parasite invasion of the mosquito’s salivary glands, a critical step to ensure transmission to a new host. The authors provide further data supporting the importance of saglin for the parasite. But their data suggest that saglin does not function as receptor during salivary gland invasion. Saglin is not expressed in those lobes of the salivary gland that are preferred by the parasites to enter the organ, which is in agreement with other cited literature.

In addition, the authors provide evidence that saglin is contained in the saliva, which is also ingested during a blood meal. Rather than being a receptor for salivary gland invasion, the presented data suggest that saglin promotes parasite development early after a blood meal.

This study is of interest for the readership of PLOS Pathogens, well-written, and the presented data are clear. Still, this manuscript may benefit restructuring to help the reader comprehending the importance of the findings. Indeed, it may be worth to consider starting the manuscript with the data on saglin (including now Fig. 8), saglin expression and localization (which support previous reports) and finishing with the new data on saglin’s pro-parasitic effect soon after the infectious blood meal.

Below are specific comments, intended to further strengthen the manuscript.

Reviewer #3: This ms continues the interest in Saglin as a mosquito molecule potentially important in the infectivity of malaria parasites to mosquitoes, although studies by other groups have strongly suggested otherwise. There is an interesting conundrum presented here, that Saglin is important (but not essential) for midgut ‘colonization’ to the oocysts stage, and thereafter it has no effect at all on the conversion of oocyst to sporozoite.

The big point here also is that the effects of Saglin are observed only when the infections in mosquitoes are low. This is based on three experimental steps, each of which are slightly flawed in their approach and leave the data open to interpretation.

1. The calculation of bloodmeal size: this is done by weighing pools of mosquitoes after feeding. However, the same mosquitoes were not weighted before feeding, so it is impossible to determine what comprises final weight due to larval nutrition (resulting in increased adult body size – see Parasites & Vectors 6, 345 (2013)) or bloodmeal size. Indeed, the mosquito size after feeding seems several orders of magnitude higher (mg rather than �g) that would be expected, and the high variation in adult size from <2 mg [sic] to >5 mg suggests that rearing conditions were far from optimal.

2. The assessment of microgametocytemia is not well controlled. The blood meals were smeared onto a slide but there is no mention of how this was standardized to give a constant number of red cells read per midgut preparation. This is important when one considers the statistical approaches to sampling. When one looks at figure 3F, the presence in mosquitoes fed on highly infected mice of more insects with no microgametes in the gut is very unexpected and unexplained. Similarly, there is no mention in the methods of how high and low falciparum infections were produced. Coupled with the issues in 1, there is a real risk of multiplication of variables to give results not associated with the biology of the system. The authors should read papers by Jeff Vaughan (especially Trends Parasitol. 23:63-70 (2007)) to understand the change in dynamics of infection at different gametocytemias in the host. The fact that macrogametes are ignored in the assessments is also a cause for come concern.

3. The high and low gametocytemias in the mice is unexplained. In the methods, all the mice were treated identically and yet produced different infection and different infectivities to the mosquitoes. Did the authors consider that other factors are at play here besides simple high and low microgametocytemias?

Put together, the authors could offer a more convincing case that their basic experimental design, on which all the molecular manipulations are overlaid, and the sampling and statistical assessments are robust.

In general, the methods are reasonably well written though I have offered many suggestions below. However, missing from the experimental design sections is information and detail concerning the sample sizes and replicates. These are hard to figure out later in the ms.

I have no comments on the genetic modifications in this ms; they are very sound and supported by strong analytical and visual data.

One major interpretation in this ms is the possible effect of Saglin as a secreted molecule being active in the midgut. In this regard the authors might have considered a simple method for assessing secreted salivary proteins (Billingsley et al. Trans RSTMH 85:450-453 (1991), and the evidence for salivary gland sporozoites and enzymes appearing in the midgut (Beier et al. AJTMH 47:195-207 (1992) and many papers by Jose Ribeiro) that actually support the physiological basis for observations.

**Part II – Major Issues: Key Experiments Required for Acceptance**

Reviewer #1: 1. The study as it stands lacks direct evidence of a role of re-ingested saliva-derived saglin in promoting Plasmodium infection in the midgut. The authors performed priming or co-feeding experiments to test this hypothesis but unfortunately no significant effect was observed in this experimental setup. The authors could attempt a more direct approach and test the effect of recombinant saglin protein supplementation in membrane feeding assays.

2. The absence of saglin results in reduced oocyst burden but it is not clear at which step saglin is acting in the midgut. Is it an activation factor for male gamete formation? Is it involved in ookinete formation or migration through the midgut epithelium? Here also the authors could use the recombinant protein to test of saglin activates gametocytes, binds to ookinetes and/or promote ookinete migration.

Reviewer #2: (No Response)

Reviewer #3: (No Response)

**Part III – Minor Issues: Editorial and Data Presentation Modifications**

Reviewer #1: -line 29: unicellular

-line 109-110: SM1 is a mimotope of TRAP and binds to saglin not to TRAP

-throughout the manuscript the authors sometimes use “distal” to refer to “distal lateral” or “lateral” when describing salivary gland lobes.

-In Fig2 (and others) the graphs show median values, while the text refers to mean values. It might be useful to harmonize the description of the data between figures and text.

-Line 430: Fig2 supplt 1 instead of Fig 2C

-Line 494: mosquitoes were

-Lines 674-678: the authors could discuss the possible reasons why the results are discordant with the Ghosh et al study. Do they envisage that TRAP-saglin interaction could still play a role in the salivary glands for transmission?

-Fig2, 3, 5, 6, 7, 2S1, 5S1: please carefully check the figures, frames are visible around some of the panels.

-line 929: why showing five countings instead of the three values for the independent experiments?

-line 930: the standard error is not shown in fig2C

-line 988: DL and PL likely refer to distal lateral and proximal lateral, respectively

Reviewer #2: A rescue experiment would certainly provide robust support for the hypothesis that “saglin ingested with the blood meal may have a pro-parasitic effect on Plasmodium midgut colonization”. But the conducted experiment (Fig. 6) seems not well-fitted to test this hypothesis. Especially since the presence of WT saliva (including saglin) at the bite site cannot be controlled for. Also do mosquitoes use the same bit site at all? A better rescue experiment may be to supplement saglin in a membrane feed into sag(-)KI mosquitoes. Thus, although it is formally not wrong, Fig. 6 and the respective text is not convincing as these experiments do not answer the question whether saglin supplementation can rescue parasite development in sag(-)KI mosquitoes. Please revise.

The experiment leading to the data shown in Fig. 5B is unclear. Please explain what XFD594 is and, more importantly, how WGA injection helps answering the question if sporozoites can access saglin coming from the hemolymph. Currently, the significance of this experiments and thus the conclusions of the WGA staining are unclear.

L 47: Please explain the abbreviation DENV as you are using it later in line 51.

L 77 and throughout the manuscript: To distinguish Anopheles and Aedes mosquitoes, I thought Anopheles is abbreviated An. Please edit accordingly, especially since Aedes aegypti in line 654 is also abbreviated as “A.” Also, it is unclear why Anopheles is sometimes abbreviated and sometimes not. Please harmonize.

L98: What is a one to one orthologue? Is it highly conserved?

L 106-109: The discussion would benefit from mentioning the injected anti saglin mAB work again and how these results fit to new proposed function of saglin.

L 109: Please provide more information on the circular peptide SM1. Where does it come from?

L 177: Should it be 2 x 10^7 rather than 2.1^7?

L 420 and following: It is confusing to report the mean in the text and the median in the figure. Although not wrong, please consider revising this.

Similarly, in the text wild-type mosquitoes are described as control, but in the Fig. wild-type is used as label.

L430: Fig. 2C shows the number of SGS not the number of oocysts. Please revise.

Caption of Fig. 2E: Please add that the red bars indicate.

A panel “F” is not part of Fig. 2 and the respective caption. Please revise.

L 463: “Plasmodia” is not a taxonomical term and should neither be italicised nor capitalized. Also, it has been critically discussed in the literature (Plasmodia - don’t by McFadden, 2012). Please change.

Throughout the manuscript: Please consistently use either wild type or WT, see e.g., in lines 466 and 469.

L 473 and 474: Please provide a reference for this statement or delete.

Fig. 3E: Could you use a scale bare that has not the approximate length of 7 µm? Also, in the other figures, the length of the scale bar is provided in the caption rather than the image itself.

L 548: Here a little more context would be helpful. To my knowledge sporozoites prefer the lateral lobes, which could be mentioned in this sentence. And you show that saglin not expressed in the lateral lobes. Just found this information much below (line 567), including a reference. Perhaps consider providing this information already here.

L 564: As there is no statistical analysis, the term “significantly” should be revised.

L 589: Please explain what proboscis samples are. Is this saliva or the actual organ?

Paragraph starting in line 618 does not reference Fig. 7B. Please revise. Also, the data shown in Fig. 7 are related to those in Figs. 2 and 3 and it is unclear why these are separated (see also comment above).

L696: “regurgitate” does not seem to make sense in this context. Should it have been re-ingest?

Fig. 8 and the corresponding text in line 732-736 should be incorporated in the results section, perhaps as new Fig. 1.

Reviewer #3: Specific comments (by line number):

59 The red cell membrane is ruptured rather than shed.

63 The language is imprecise here. The ookinetes traverse the midgut epithelial cells, not the whole epithelium. The ookinetes rest upon the basal lamina not below it. The cells confer a polarity to the basal lamina, and the ookinetes rest on it and facing the haemocoele but still separated from it by the basal lamina.

66 The ookinete is not half-moon shaped (that would be a semi-circle).

69 Should read kinetics.

70 Most of the introduction to this point is superfluous – does not contribute to the core of the ms.

135 Is the key resources table required by the journal? It seems an odd inclusion. It does not have a legend and references should go back to source (for example NF54 was generated in The Netherlands many years prior to Bryant et al. 2018).

139-42 The mosquito strains here do not appear in the resources table

148 and elsewhere – “h” is the correct abbreviation for hours

149 “Osmosed” is not correct

154 why “floated”?

160 change “neonate” (a term describing a mammalian baby) to “first instar”

172 Again, precision – the mice were injected with thawed P. berghei blood stages in mouse blood….how many parasites per mouse? How controlled were these infections? Were mice treated beforehand with phenylhydrazine as done in all Sinden papers?

179 delete “when”

185 Bryant is an inappropriate reference here

189-90 Number “2” should be subscripted in chemical formulae

195 parafilm should be Parafilm™

203 Usually berghei oocysts are counted on day 10, falciparum on day 7-8

208 replace ,, with “

212 Oocysts were manually counted using software? I don’t understand.

215 Ground not grinded; here and throughout ms, use decimal point rather than comma (1.5 mL)

219 Why such low magnification? There would be a lot of detritus here which could easily obfuscate the counting

245-6 Not clear if individuals were weighed or pools

247 Their abdomens “were”…

249 7 �l “were” loaded (please check throughout). How did the authors control for rupture of red cells using this method?

Note – I have stopped going through the methods line-by-line here – it is too much work for a reviewer

PLOS authors have the option to publish the peer review history of their article (what does this mean?). If published, this will include your full peer review and any attached files.

Reviewer #1: No

Reviewer #2: No

Reviewer #3: No
---

## [Decision Letter · Decision Letter 1]

16 Jan 2023

Dear Mr. Klug,

Thank you very much for submitting your manuscript "The salivary protein Saglin facilitates efficient midgut colonization of Anopheles mosquitoes by malaria parasites" for consideration at PLOS Pathogens. As with all papers reviewed by the journal, your manuscript was reviewed by members of the editorial board and by several independent reviewers. The reviewers appreciated the attention to an important topic. Based on the reviews, we are likely to accept this manuscript for publication, providing that you modify the manuscript according to the review recommendations.

Sincerely,

Oliver Billker

Academic Editor

PLOS Pathogens

Kirk Deitsch

Section Editor

PLOS Pathogens

Kasturi Haldar

Editor-in-Chief

PLOS Pathogens

orcid.org/0000-0001-5065-158X

Michael Malim

Editor-in-Chief

PLOS Pathogens

orcid.org/0000-0002-7699-2064

Reviewer Comments (if any, and for reference):

Reviewer's Responses to Questions

**Part I - Summary**

Reviewer #1: (No Response)

Reviewer #3: The authors have undertaken a lot of work to address what were substantial comments from three referees. While some of the results are still not conclusive, further work would be outside the scope of this manuscript.

**Part II – Major Issues: Key Experiments Required for Acceptance**

Reviewer #1: (No Response)

Reviewer #3: None.

**Part III – Minor Issues: Editorial and Data Presentation Modifications**

Reviewer #1: The authors have modified the text to better reflect the existing data. While it is unfortunate that they did not attempt the rescue experiment through membrane feeding (which would probably be more efficient than injection into mice), they do refer to this experiment in the discussion. It will be interesting to know the outcome of such an experiment, which could bring some novel insights into the role of saglin during Plasmodium infection.

I only have a few minor comments:

1) Fig1B could be cited line 130

2) Fig3C shows no difference in weight after blood meal, therefore the statement line 502-503 should be modified.

3) The western blot in fig S5 is intriguing, as there is much more saglin detected in 5 midgut samples as compared to 8 salivary glands. It looks like there is a massive protein band (presumably albumin) in the WT lane that could interfere with saglin detection. For some reason this interfering protein accumulation is not seen (or not to the same extent) in the KO lane. I suggest removing this figure. The information that saglin is detected in the midgut of blood-fed mosquitoes is already shown in fig 5F.

Reviewer #3: (No Response)

PLOS authors have the option to publish the peer review history of their article (what does this mean?). If published, this will include your full peer review and any attached files.

Reviewer #1: No

Reviewer #3: **Yes: **Peter F Billingsley

Figure Files:

Data Requirements:

Reproducibility:

References:

---

## [Editor Report · Decision Letter 2]

8 Feb 2023

Dear Mr. Klug,

We are pleased to inform you that your manuscript 'The salivary protein Saglin facilitates efficient midgut colonization of Anopheles mosquitoes by malaria parasites' has been provisionally accepted for publication in PLOS Pathogens.

Best regards,

Oliver Billker

Academic Editor

PLOS Pathogens

Kirk Deitsch

Section Editor

PLOS Pathogens

Kasturi Haldar

Editor-in-Chief

PLOS Pathogens

orcid.org/0000-0001-5065-158X

Michael Malim

Editor-in-Chief

PLOS Pathogens

orcid.org/0000-0002-7699-2064
---

## [Editor Report · Acceptance letter]

24 Feb 2023

Dear Mr. Klug,

We are delighted to inform you that your manuscript, "The salivary protein Saglin facilitates efficient midgut colonization of Anopheles mosquitoes by malaria parasites," has been formally accepted for publication in PLOS Pathogens.

Best regards,

Kasturi Haldar

Editor-in-Chief

PLOS Pathogens

orcid.org/0000-0001-5065-158X

Michael Malim

Editor-in-Chief

PLOS Pathogens

orcid.org/0000-0002-7699-2064